# Deconstructing Positional Information: From Attention Logits to Training Biases

**Zihan Gu**[1,2], **Ruoyu Chen**[1,2], **Han Zhang**[3], **Hua Zhang**[1,2,*], **Yue Hu**[1,2]

[1]Institute of Information Engineering, Chinese Academy of Sciences;

[2]School of Cyber Security, University of Chinese Academy of Sciences;

[3]Antai College of Economics and Management, Shanghai Jiao Tong University.

`{guzihan,chenruoyu,zhanghua,huyue}@iie.ac.cn`

## Abstract

Positional encodings enable Transformers to incorporate sequential information, yet their theoretical understanding remains limited to two properties: distance attenuation and translation invariance. Because natural language lacks purely positional data, the interplay between positional and semantic information is still underexplored. We address this gap by deconstructing the attention-logit computation and providing a structured analysis of positional encodings, categorizing them into additive and multiplicative forms. The differing properties of these forms lead to distinct mechanisms for capturing positional information. To probe this difference, we design a synthetic task that explicitly requires strong integration of positional and semantic cues. As predicted, multiplicative encodings achieve a clear performance advantage on this task. Moreover, our evaluation reveals a hidden training bias: an information aggregation effect in shallow layers that we term the single-head deposit pattern. Through ablation studies and theoretical analysis, we proved that this phenomenon is inherent in multiplicative encodings. These findings deepen the understanding of positional encodings and call for further study of their training dynamics. Our code is available at `https://github.com/Qihuai27/Deposit-Pattern-Research`.

## 1 Introduction

Positional Encoding (PE) (Vaswani et al., 2017; Su et al., 2024) endows Transformer architectures with an understanding of sequence order, compensating for the permutation-invariant nature of self-attention (Vaswani et al., 2017). This component is critical to their success in both language (Vaswani et al., 2017) and vision (Dosovitskiy et al., 2021) tasks. The design of PE has evolved substantially, from the additive sinusoidal embeddings (Vaswani et al., 2017) of the original Transformer to trainable absolute embeddings (Shaw et al., 2018) and more sophisticated relative schemes like T5 biases (Raffel et al., 2020) and Rotary Positional Encoding (RoPE) (Su et al., 2024). This progression reflects a sustained effort to enhance generalization, extend context length, and improve model efficiency.

Different positional encodings model distance decay and translation invariance in different ways, but these differences are difficult to observe in the face of complex data with human grammatical rules. Current evaluation paradigms, which rely on aggregate metrics such as perplexity (Ermo et al., 2025) or length-extrapolation benchmarks (Kazemnejad et al., 2023), document these performance differences but fail to illuminate the underlying mechanisms of how positional signals are processed (Peng et al., 2024; Ermo et al., 2025). Therefore, theoretical research on positional encoding relies on synthesis tasks (Zuo et al., 2025; Kazemnejad et al., 2023), but some unexplained phenomena have emerged. For example, despite its strong theoretical properties, such as the attenuation characteristic that supports length generalization, RoPE can underperform simpler relative PEs or even models with no PE on certain tasks (Kazemnejad et al., 2023). This discrepancy highlights a critical gap in our understanding: ***How, precisely, do different PE schemes mediate the interaction between***

---

*Corresponding Author

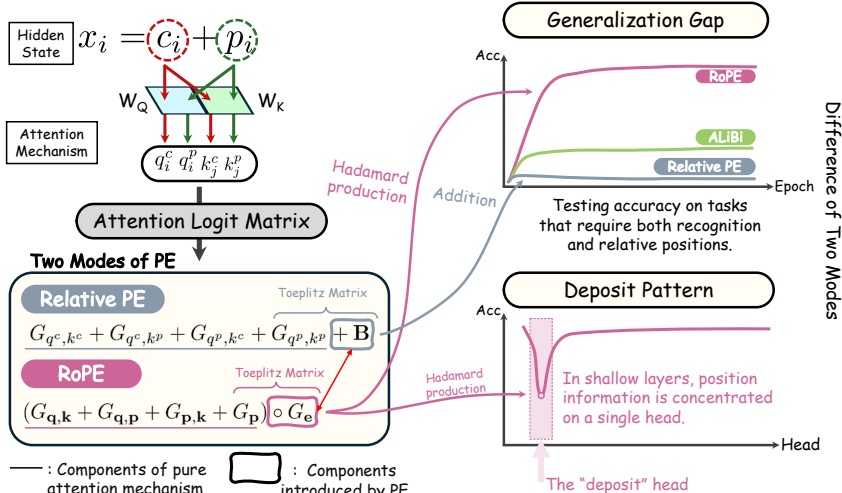

Figure 1: **Schematic Overview. (Left)** Each token is viewed as $x_i = c_i + p_i$. After the $W_Q, W_K$ projections, their interactions form the four inner-product terms of the logit matrix. The framework unifies PE methods by how they introduce Toeplitz (translation-invariant) structure: additive PEs (Absolute, T5, ALiBi) contribute a Toeplitz bias $B$ or a Toeplitz $G_{q^p, k^p}$; RoPE applies a relative-position Toeplitz kernel $G_e$ multiplicatively through a Hadamard product. **(Right)** These two modes produce distinct behaviors: RoPE's multiplicative coupling gives faster learning and smaller generalization gaps, yet concentrates shallow-layer positional information into a "deposit head".

***token content and position?*** This lack of mechanistic clarity hinders both the explanation of existing performance trade-offs and the principled design of future PE schemes.

To bridge this gap, our approach is grounded in a core principle of positional encoding: the relationship between any two tokens should depend on their **relative distance**, not their absolute positions. This property of translation invariance is formally captured by the structure of a **Toeplitz matrix**, in which all elements along any given diagonal are identical. This insight allows us to deconstruct the attention logit calculation and propose a unified, Toeplitz-based framework for analyzing PE mechanisms (Figure 1). In our framework, each token's embedding is composed of a **content component** (what it is) and a **position component** (where it is). The central idea is that these position components collectively induce a Toeplitz structure in the final attention scores.

This framework clearly distinguishes between two primary classes of PE. **Additive methods**, such as relative position biases, operate by adding a static Toeplitz matrix of position scores directly to the attention logits. In contrast, **multiplicative methods** like RoPE dynamically integrate positional information into the query and key representations. This creates a powerful **content–position interaction**, where the positional influence on attention scores is conditioned on the token's content. While this coupling provides a significant advantage on tasks where meaning is tightly bound to relative location, we argue that it also introduces a strong inductive bias. This bias can manifest as a drawback, concentrating the learning of positional logic into a small, specialized subset of attention heads.

Observations based on the theoretical framework directly triggered our experimental design: we conduct experiments on a series of carefully designed synthetic tasks. We generate random sequences from a small vocabulary and use labeled "anchor" tokens to define two contrasting objectives: a **position-sensitive task** requiring reasoning about the relative positions of anchors, and a **position-agnostic task** dependent only on token counts. This controlled setup enables us to precisely isolate and evaluate the model's positional reasoning capabilities. Our experiments confirm that RoPE decisively outperforms other methods on the position-sensitive task while underperforming on the position-agnostic one. More importantly, this setting reveals a striking artifact unique to RoPE: the **"single-head deposit pattern"**, where nearly all positional processing in the shallow layers becomes concentrated into a single attention head.

Further investigation confirms this phenomenon is unique to RoPE on tasks with strong content-position correlation. Through rigorous ablation studies and theoretical derivation—such as injecting absolute positional encodings or selectively applying RoPE to a subset of heads—we demonstrate that the deposit pattern is an **intrinsic property** of RoPE's multiplicative structure, not an incidental training artifact. This finding also explains why Transformers using RoPE tend not to form the latent

positional representations observed in models with additive or no PE (Zuo et al., 2025; Kazemnejad et al., 2023). We posit that this intense, structurally-induced specialization is a primary cause of the gap between RoPE's theoretical promise and its practical performance.

Our contributions can be summarized as follows:

- We propose a unified analytical framework using Toeplitz matrices that categorizes positional encodings into additive and multiplicative mechanisms, clarifying their distinct effects on attention logits.
- We empirically identify and analyze RoPE's performance paradox through targeted synthetic tasks, revealing a unique 'single-head deposit pattern' where positional logic becomes highly concentrated in shallow layers.
- We conduct a causal analysis to demonstrate that the deposit pattern is an intrinsic property of RoPE's multiplicative architecture, offering a mechanistic explanation for its observed performance paradox.

## 2    RELATED WORK

**Positional Encoding in Transformers.** Positional encoding was introduced alongside the Transformer architecture to endow self-attention with sequence order information (Vaswani et al., 2017). Early approaches employed learnable positional embeddings (Shaw et al., 2018), while subsequent work integrated position biases directly into the attention logits, giving rise to relative position encoding and its extensions (Press et al., 2022; Li et al., 2024; Chi et al., 2023; 2022; Raffel et al., 2020; Su et al., 2024; Zhao et al., 2025). Among these, Rotary Positional Encoding (RoPE) (Su et al., 2024) stands out by applying a multiplicative bias to the $QK^\top$ matrix, and has been widely adopted in modern open-source models. Related efforts on sliding-window and length-generalization designs are also often framed as positional encoding variants (Kiyono et al., 2021; Chen et al., 2024a;b; Ding et al., 2024; Peng et al., 2024). Recent RoPE-based improvements include Frequency-Partitioned Encoding (FoPE) (Ermo et al., 2025) and the Multi-head Latent Attention (MLA) mechanism in Deepseek-V3 (Liu et al., 2024).

**Mechanistic Analyses of Positional Encoding.** Empirical studies have probed how Transformers internalize positional signals. Some works demonstrate that even without explicit positional embeddings (NoPE), models can learn positional distinctions (Wang et al., 2024; Zuo et al., 2025; Haviv et al., 2022; Barbero et al., 2024), occasionally outperforming PE-equipped counterparts on select tasks (Kazemnejad et al., 2023). Explanations range from implicit causal masking (Zuo et al., 2025) to dataset-dependent effects (Barbero et al., 2024). Other analyses document characteristic behaviors of RoPE, such as "massive value" phenomena and rotation artifacts (Jin et al., 2025; Jonasson, 2025).

**Toeplitz Structure and Spectral Theory.** The action of positional encoding in attention can be unified under Toeplitz-matrix representations (Böttcher & Silbermann, 2006; Grenander, 1958; Gray, 2006; Oppenheim & Schafer, 1999). Classical results on Toeplitz spectral distributions, including Szegő's theorem, inform our theoretical framework. Moreover, the impact of matrix spectra on optimization convergence is well studied in numerical linear algebra and convex optimization contexts (Horn & Johnson, 2012; 1994; Boyd & Vandenberghe, 2004; Nesterov, 2013).

## 3    METHODOLOGY

### 3.1    PRELIMINARIES: POSITIONAL ENCODING IN ATTENTION

The Transformer attention score between a query $q_i$ at position $i$ and a key $k_j$ at position $j$ is based on their inner product, $q_i^\top k_j$. Since this computation does not depend on sequence order, Positional Encoding (PE) schemes introduce positional information by modifying either the token representations or the attention logits.

**Absolute PE** (Vaswani et al., 2017) adds a position vector $p_i$ to the initial embedding $e_i$:

$$h_i = e_i + p_i.$$

The query and key vectors derived from $h_i$ therefore mix content and position inside the representation, making the attention score depend on the *absolute* token positions.

**Relative PE** (Raffel et al., 2020) incorporates positional information directly into the attention logits through a learnable distance-dependent term:

$$\alpha_{i,j} \propto \exp\left(q_i^\top k_j + b_{j-i}\right).$$

A defining feature of relative PE is *translation invariance*: the positional contribution depends only on the displacement $j - i$, not on the absolute locations of $i$ and $j$. Within this family, T5 (Raffel et al., 2020) uses a learned table indexed by $(j - i)$, while ALiBi (Press et al., 2022) applies a fixed slope that increases linearly with distance. Thus both are translation-invariant, but differ in whether the distance mapping is learned (T5) or predetermined (ALiBi).

**Rotary PE (RoPE)** (Su et al., 2024) introduces position by rotating the query and key vectors with a position-dependent matrix $R_i$. A key identity shows that the resulting inner product depends only on the relative displacement:

$$(R_i q_i)^\top (R_j k_j) = q_i^\top R_{j-i}^\top k_j.$$

Thus RoPE implements positional encoding through a structured rotation that couples content and position via relative position, rather than through additive shifts or explicit bias terms.

## 3.2 A Unified Framework via Toeplitz Matrices

To analyze these mechanisms within a common framework, we introduce two foundational assumptions. The first is a conceptual decomposition of token representations.

**Assumption 3.1** (Representation Decomposition). Any token representation $x_i$ can be conceptually decomposed into a position-independent **content component** $c_i$ and a position-dependent **position component** $p_i$, such that $x_i = c_i + p_i$.

*Remark* 3.1. This decomposition is not a modeling constraint but a structural lens. It is supported in four complementary ways: **(i) Design:** Absolute PE explicitly defines position-dependent vectors $p_i$ (Vaswani et al., 2017); **(ii) Emergence:** Architectures lacking PE still develop internal positional directions (Zuo et al., 2025); **(iii) Functionality:** Ablating $p_i$ hurts performance more than ablating random vectors of equal norm (Section 5.4); **(iv) Generality:** We treat the removal of $p_i$ as a manifold-based coordinate subtraction rather than a rigid linear projection, ensuring compatibility with diverse PE schemes.

This allows us to analyze how PE schemes construct or interact with the positional component $p_i$. The core principle uniting relative PE schemes is **translation invariance**: the positional component of the attention score depends only on the relative distance $j - i$. This principle is perfectly embodied by a specific matrix structure.

**Definition 3.1** (Toeplitz Matrix). A matrix $\mathbf{T}$ is a Toeplitz matrix if its entries are constant along each diagonal, i.e., $T_{i,j} = a_{i-j}$ for some sequence $\{a_k\}$. It is defined by $2N - 1$ unique values for an $N \times N$ matrix.

Such matrices naturally arise whenever a quantity depends only on the relative displacement $j - i$, making them the canonical representation of translation-invariant positional interactions. This leads to our second assumption, which connects the positional components of tokens to the structure of the attention matrix.

**Assumption 3.2** (Toeplitz Structure from Positional Interaction). Since the positional contribution to attention depends only on the relative displacement $j - i$, the Gram matrix formed by the position-dependent components naturally takes a Toeplitz form.

*Remark* 3.2. Assumption 3.2 can be viewed less as a modeling assumption and more as a working hypothesis that captures the intended translation-invariant decay, and that the $p_i$ decomposed from Assumption 3.1 should have such a property. This assumption could be checked by sinusoidal absolute PEs and is explicitly enforced by construction in relative PE biases. It provides a powerful analytical tool for understanding the structure of positional information in the attention matrix.

Under Assumptions 3.1–3.2, the attention logits can be decomposed into interpretable content and position interactions. Let $q_i^c, k_j^c$ be query/key vectors derived from content components and $q_i^p, k_j^p$ be those from position components. Let $G_{v,w}$ denote the Gram matrix where $(G_{v,w})_{i,j} = v_i^\top w_j$.

For **additive mechanisms** (e.g., Absolute or Relative PE), the logit matrix is a sum of components:

$$\mathbf{L}_{\text{Additive}} = G_{q^c, k^c} + G_{q^c, k^p} + G_{q^p, k^c} + G_{q^p, k^p} + \mathbf{B} \tag{1}$$

Here, $\mathbf{B}$ denotes the explicit relative-bias matrix used only by relative encodings such as T5 and ALiBi. For absolute or learned absolute PEs, we instead have $\mathbf{B} = 0$, and the Toeplitz positional structure is provided solely by the term $G_{q^p, k^p}$. Thus, in all additive mechanisms the Toeplitz component arises either from $G_{q^p, k^p}$ (absolute PE) or from $\mathbf{B}$ (relative PE), while the cross-terms $G_{q^c, k^p} + G_{q^p, k^c}$ remain the only channel through which content can interact with relative position.

In contrast, **multiplicative mechanisms** like RoPE induce a different structure. Following its formulation using complex numbers in Su et al. (2024), the logit matrix becomes:

$$\mathbf{L}_{\text{RoPE}} = \text{Re}\left\{ (G_{q^c, k^c} + G_{q^c, k^p} + G_{q^p, k^c} + G_{q^p, k^p}) \circ G_{\mathbf{e}} \right\} \tag{2}$$

where $\circ$ is the Hadamard product and $G_{\mathbf{e}}$ is a Toeplitz matrix.

*Remark* 3.3. The complex form of RoPE is an important component of the original paper (Su et al., 2024). This form, combined with the Abel summation, can prove an upper bound on its distance decay. A detailed derivation of how to convert the complex RoPE form into the Toeplitz matrix form above is provided in Appendix B. This formulation highlights that RoPE does not add position information; instead, it *modulates* all content interactions through a shared position-dependent kernel.

This formal analysis reveals a critical distinction. Additive methods combine content and position signals, but RoPE's multiplicative structure uses a Toeplitz kernel ($G_{\mathbf{e}}$) to modulate the *entire* content-content and content-position interaction. This enables a more direct and expressive coupling, allowing the model to learn attention patterns where the relevance of a token's content is conditioned on its relative position to another. This type of task cannot be achieved through pure translation invariance; instead, it requires that a specific translation distance be directly affected by the content feedback. We hypothesize that this powerful coupling mechanism comes with a drawback: by routing the learning of positional dependencies through this multiplicative kernel, RoPE may create a strong inductive bias that encourages **positional specialization**, concentrating this logic into a small subset of attention heads. This hypothesis motivates the experiments in the following section.

## 4 EXPERIMENTS

To empirically test our hypothesis about the distinct behaviors of additive and multiplicative positional encodings, we design two synthetic tasks. These tasks allow us to isolate the model's ability to couple token content with positional information in a controlled setting. We use a 6-layer Transformer decoder as our base model and evaluate six PE configurations: Absolute (Vaswani et al., 2017), T5 Relative Bias (Raffel et al., 2020), ALiBi (Press et al., 2022), RoPE (Su et al., 2024), NoPE (no explicit PE) (Kazemnejad et al., 2023), and a Random initialized embedding baseline. Our experiments are averaged over five seeds, and the variance is less than 5%. Full implementation and training details are provided in Appendix D.

### 4.1 SYNTHETIC TASK DESIGN

**Task 1 – Position-Sensitive: Relative Distance Classification.** This task is designed to require strong content-position coupling, the theorized strength of multiplicative mechanisms. Each input sequence contains two unique "trigger" words drawn from a vocabulary of 1,000 tokens. The model must predict the relative distance between them, formulated as a classification problem over binned distances. Success requires the model to identify *what* the trigger words are and determine *where* they are in relation to each other. We generated a dataset of 100,000 examples for this task.

**Task 2 – Position-Agnostic: Trigger Word Counting.** This task serves as a control, designed to penalize any inflexible positional bias. The model must predict the total number of occurrences of a predefined set of 20 trigger words within a sequence, regardless of their positions. Because positional information is a nuisance variable, an ideal model should learn to ignore it. We generated a dataset of 50,000 examples for this task.

## 4.2 Performance and Generalization Analysis

We first analyze the learning dynamics and generalization performance of each PE method on our two tasks. The results are presented in Figures 2 and 3.

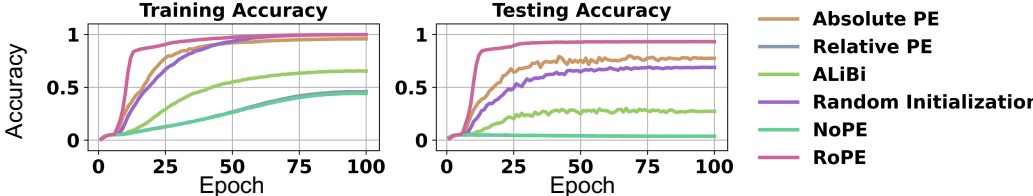

Figure 2: Training and test accuracy on **Task 1 (Position-Sensitive)**. RoPE's multiplicative coupling enables it to significantly outperform all other methods in both convergence speed and final generalization.

**On the Position-Sensitive Task, RoPE Excels.** As predicted by our framework, RoPE (Figure 2) demonstrates superior performance. It converges fastest and achieves the highest test accuracy with a minimal generalization gap. This supports our theory that its multiplicative content-position coupling is uniquely suited for tasks demanding this form of reasoning. In contrast, additive methods show mixed results. Absolute PE performs second best, as its initial positional signal can be adapted by subsequent layers. The Random embedding baseline learns the training set but fails to generalize, as its non-Toeplitz structure lacks the translation invariance required to learn a general rule. Methods like ALiBi, with its fixed and data-agnostic Toeplitz bias, struggle to adapt, while NoPE and standard Relative PE fail entirely, lacking any mechanism to bind content to position.

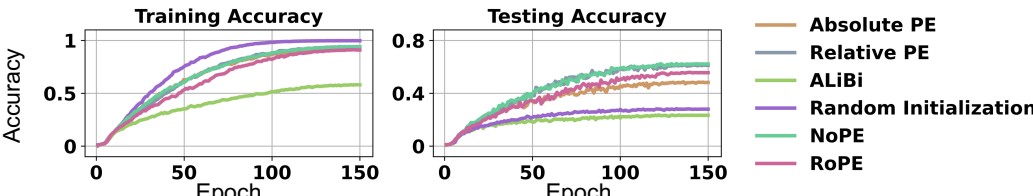

Figure 3: Training and test accuracy on **Task 2 (Position-Agnostic)**. Strong, inflexible positional biases like ALiBi are detrimental. RoPE's adaptive nature allows it to perform well, nearly matching methods with no positional bias.

**On the Position-Agnostic Task, Strong Positional Biases are Detrimental.** Figure 3 confirms our second prediction: on a task where position is irrelevant, a strong positional inductive bias is harmful. ALiBi, with its rigid, non-adaptive bias, suffers the most severe performance degradation. Conversely, NoPE and Relative PE (which learns to attenuate its bias towards zero) perform well, as they do not impose a counterproductive positional structure. RoPE also performs strongly, demonstrating its adaptability; its multiplicative mechanism allows the model to learn to down-weight the positional signal when it is not useful. Absolute PE incurs a small penalty, while the Random baseline again overfits, learning spurious correlations between positions and counts in the training data.

## 4.3 Evidence of the Single-Head Deposit Pattern in RoPE

Our theory suggests that RoPE's multiplicative structure concentrates positional reasoning into a few specialized heads. To find direct evidence for this phenomenon, we conduct a head-wise causal ablation study on the models trained on Task 1. For each head in the network, we zero out its output and measure the resulting drop in test accuracy. A large drop indicates that the head is critical to the task.

The results, shown in Figure 4, provide striking confirmation of our hypothesis. **In the RoPE model, nearly all positional logic is localized to a single head in the first layer.** Ablating this one head causes a catastrophic drop in accuracy ($\approx$60%), while ablating other heads has a negligible effect. We term this phenomenon the **"single-head deposit pattern."** Crucially, this pattern is unique to the combination of the RoPE architecture and a position-sensitive task. As shown in Figure 4, the pattern does not emerge in the NoPE model on the same task, nor does it appear in the RoPE model

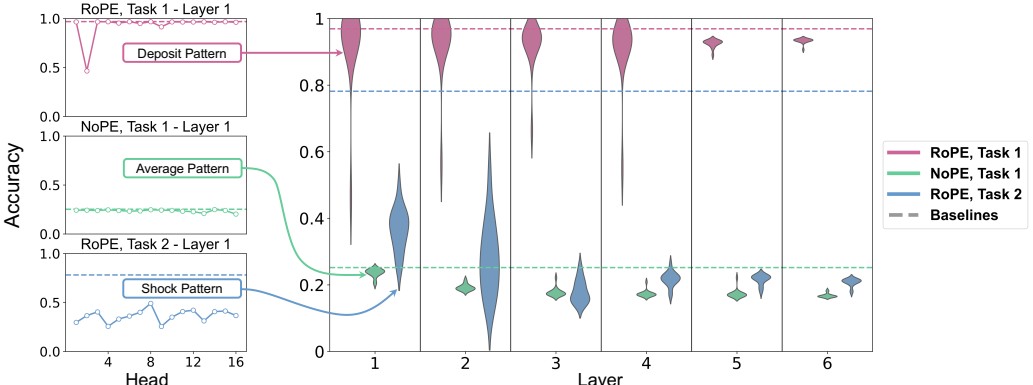

Figure 4: **Head-wise ablation reveals a unique "Deposit Pattern" in RoPE.** Each violin shows the distribution of accuracies after ablating each head in that layer; long vertical tails correspond to outlier heads whose removal causes a large accuracy drop. The three insets illustrate the characteristic shapes of these distributions: a "deposit" shape with a single deep outlier (RoPE–Task 1), a flat "average" shape (NoPE–Task 1), and a noisy "shock" shape (RoPE–Task 2). These shapes match the contours of the main violins. Full per-head curves are provided in Appendix D.2.

trained on the position-agnostic Task 2. This confirms that the deposit pattern is not a generic artifact of Transformer training but a direct consequence of RoPE's multiplicative content-position coupling mechanism. We therefore interpret the deposit pattern as a training bias rather than clean modularity, although a full causal link to length generalization is left for future work (Appendix E).

## 5 ABLATION STUDIES AND MECHANISTIC INSIGHTS

The discovery of the "single-head deposit pattern" raises critical questions about the underlying mechanics of RoPE. Is this pattern an inevitable side effect of multiplicative coupling, an efficient specialization, or a wasteful bottleneck? To elucidate the mechanisms driving this phenomenon, we design three targeted ablation studies, each testing a specific hypothesis.

### 5.1 ABLATION 1: DOES ROPE SUPPRESS LATENT POSITIONAL REPRESENTATIONS?

**Hypothesis:** We hypothesize that RoPE's explicit, multiplicative positional signal is so potent that it inhibits the model from forming its own implicit positional representations—a known capability of standard Transformers (Zuo et al., 2025; Kazemnejad et al., 2023)—thereby causing other heads to remain position-agnostic.

**Experimental Design:** To test this, we inject a redundant signal by augmenting a RoPE-equipped Transformer with an additional Absolute PE at the input layer. This directly provides an explicit $p_i$ component before the RoPE-enabled attention layers. We then observe if this alters the deposit pattern.

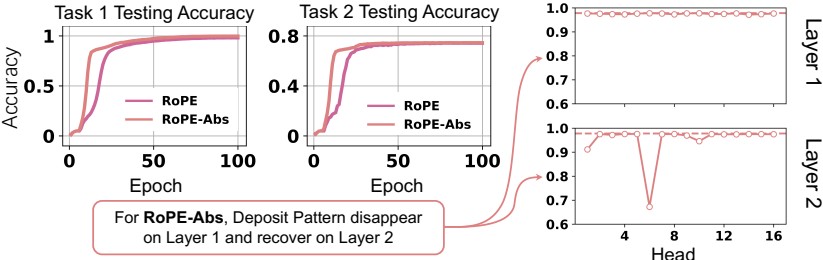

Figure 5: **(Left)** Performance of a hybrid RoPE + Absolute PE model. **(Right)** The deposit pattern is initially suppressed in Layer 1 but re-emerges in deeper layers, demonstrating the strong inductive bias of RoPE.

**Results:** The results, shown in Figure 5, are revealing. The explicit Absolute PE signal disrupts the deposit pattern in the first layer, distributing positional responsibility more evenly. However, the pattern **re-emerges** in deeper layers. This strongly suggests that while an initial explicit signal can be utilized, the powerful inductive bias of RoPE's multiplicative coupling eventually reasserts itself,

overriding the additive signal. This confirms our hypothesis: RoPE's mechanism is so dominant that it suppresses the model's tendency to form or utilize other forms of positional representation.

## 5.2 ABLATION 2: IS A SINGLE ROPE-ENABLED HEAD SUFFICIENT?

**Hypothesis:** The deposit pattern implies that for tasks like ours, a single RoPE-enabled head is sufficient for positional reasoning, rendering additional RoPE heads redundant.

**Experimental Design:** We test this by systematically reducing the number of attention heads that utilize RoPE, from all heads down to just one, with the rest operating as NoPE heads. We evaluate performance on Task 1.

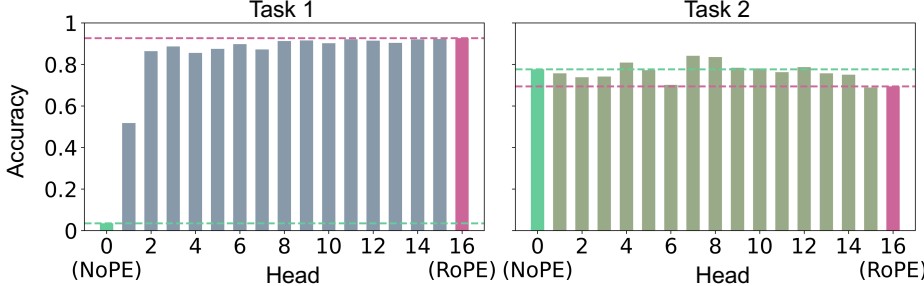

Figure 6: Performance on Task 1 when RoPE is applied to a diminishing subset of heads. Performance remains near-optimal even with just one or two RoPE-enabled heads.

**Results:** As shown in Figure 6, performance on Task 1 remains near-optimal even when only one or two heads are equipped with RoPE. This confirms that a small number of specialized heads are indeed sufficient to capture the necessary positional information for this task. The result supports our explanation for the deposit pattern: once one head effectively learns the content-position interaction via the multiplicative mechanism, other heads become inert with respect to this task, as there is no learning incentive to develop redundant positional capabilities.

## 5.3 ABLATION 3: CAN A HYBRID ARCHITECTURE MITIGATE INEFFICIENCY?

**Hypothesis:** The deposit pattern, while effective, is an inefficient use of model capacity. We hypothesize that a hybrid architecture can mitigate this by explicitly dedicating resources, retaining RoPE's strengths while improving robustness.

**Experimental Design:** To explore this, we evaluate the **Multi-Latent Attention (MLA)** architecture from DeepSeek (Liu et al., 2024). MLA formalizes a hybrid approach by creating parallel pathways for position-agnostic and position-sensitive processing within a single head. Formally, the query-key inner product is structured as a concatenation of a NoPE component and a RoPE component:

$$\langle q_i, k_j \rangle = \langle (W_{UQ}x_i \,;\, R_i W_{UR}x_i), (W_{UK}x_j \,;\, R_j W_{UR}x_j) \rangle, \tag{3}$$

where ( ; ) denotes vector concatenation and the projection matrices for each pathway are distinct. This structure ensures that each attention head computes both a NoPE-style and a RoPE-style logit simultaneously. Interpreted through our theoretical framework, this mechanism effectively modifies the logit matrix by creating a weighted combination of a standard content matrix and a RoPE-modulated one:

$$\mathbf{L}_{\text{MLA\_complex}} = (G_{\mathbf{q},\mathbf{k}} + G_{\mathbf{q},\mathbf{p}} + G_{\mathbf{p},\mathbf{k}} + G_{\mathbf{p}}) \circ (\alpha G_{\mathbf{e}} + I). \tag{4}$$

This prevents the purely multiplicative signal from dominating and encourages a more distributed representation of position. We replaced our standard attention with the MLA module and evaluated it on both tasks.

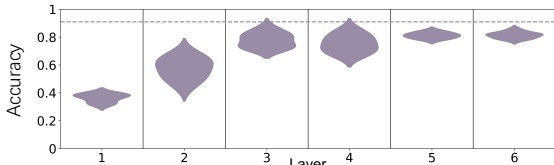

Figure 7: Head-wise ablation violin plot under MLA.

| Method | Task 1 | Task 2 |
|--------|--------|--------|
| MLA | 88.34 | 97.41 |
| RoPE | 92.64 | 69.43 |
| NoPE | 3.51 | 77.69 |

Table 1: MLA performance comparison on the two tasks.

**Results:** The MLA architecture successfully mitigates the deposit pattern. Figure 7 shows that no single head is indispensable, and positional responsibility is diffused across the model. Furthermore, Table 1 shows that MLA nearly matches RoPE's performance on Task 1 while dramatically improving on Task 2. This demonstrates that by preventing over-specialization, the hybrid model achieves a more robust balance.

### 5.4 ABLATION 4: WHERE DO ADDITIVE AND MULTIPLICATIVE PEs STORE POSITIONAL INFORMATION?

**Hypothesis.** Given the decomposition $x_i = c_i + p_i$ (Assumption 3.1), we ask whether different PE mechanisms retain an explicit positional direction $p_i$ in the activations, or whether this information is absorbed into the query–key parameterization. Our hypothesis is that additive PEs preserve activation-level $p_i$, while RoPE rapidly suppresses it.

**Experimental Design.** For each layer $\ell$, we ablate either (1) the fixed positional vectors $p_i$ (Absolute PE) or (2) norm-matched random vectors, and measure test accuracy. A larger drop under $p_i$-removal indicates that the model relies on an explicit positional direction. For RoPE, we repeat the same intervention on the hybrid Absolute+RoPE model (Section 5.1), where $p_i$ is added only at the embedding layer.

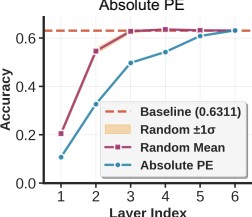 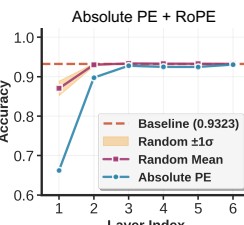

Figure 8: Layer-wise ablation of absolute positional vectors versus norm-matched random vectors. Left: Absolute PE shows consistently larger drops under $p_i$-removal, confirming that additive PEs preserve explicit positional directions in the activations. Right: For Absolute+RoPE, this difference vanishes after Layer 2, indicating that RoPE suppresses activation-level positional directions and shifts positional structure into the attention parameterization.

**Results.** Fig. 8 shows a clear contrast: (1) For Absolute PE, removing $p_i$ causes a consistently larger accuracy drop than removing norm-matched random vectors, indicating that additive PEs preserve explicit positional directions in the activations; (2) For Absolute+RoPE, this difference disappears after Layer 2, showing that RoPE suppresses activation-level positional directions and shifts positional information into the query–key parameterization.

**Summary.** Additive PEs store position in explicit activation directions $p_i$, whereas RoPE rapidly eliminates these directions and embeds positional structure in the frequency-modulated parameters. This is why the structure of MLA can eliminate this phenomenon.

## 6 THEORETICAL ANALYSIS OF THE DEPOSIT PATTERN

The empirical discovery of the "single-head deposit pattern" in RoPE (Su et al., 2024) invites a formal theoretical explanation. In this section, we provide a mathematical argument demonstrating that this phenomenon is an inherent consequence of RoPE's design. We analyze the gradient signal for the relative distance classification task (Task 1). Let $(i_s, j_s)$ be the positions of the trigger words for a sample $s$, with the task being to predict the relative distance $d_s := |j_s - i_s|$. The algebraic intuition behind our proof is this: for a sample labeled $|i - j|$, the optimization rewards the distance signal $|i - j|$ and penalizes other distance signals. For addition, the reward and penalty are always antagonistic because the position bias lacks context information. However, for multiplication, the penalty is attenuated by frequency, making the reward cumulative. This accumulation ultimately leads to an initial top-level bias in the gradient signal, which is amplified by backpropagation. As a comparison with RoPE, we also analyze ALiBi (Press et al., 2022) on this task.

Our analysis relies on the following idealized assumptions, which are empirically plausible in the context of early training.

**Assumption 6.1** (Formal Setup for Gradient Analysis).

**(A1) Jacobian Stability.** Each inter-layer Jacobian $J^{(l)}$ has singular values bounded in $[1 - \alpha_l, 1 + \alpha_l]$ for $\alpha_l \in (0, 1)$, consistent with architectures using pre-normalization.

**(A2) Monotone Anchor Path.** For each sample $s$, there is an "anchor" row $u_s$ where the gradient signal is positively correlated with the target token's value vector. Formally, $\langle \partial \ell_s / \partial Y_{u_s,:}^{(h,L)}, V_{j_s,:}^{(h,L)} \rangle \geq \eta_s^{(h)} > 0$. This formalizes that a basic learning signal exists.

**(A3) Projection Separability (for ALiBi).** The projection matrices $W_Q, W_K, W_V$ are such that $\ker W_Q \cap \ker W_K \not\subset \ker W_V$. This is a generic condition in overparameterized models.

Our analysis demonstrates a fundamental difference in how RoPE and ALiBi structure the gradient signal for relative position tasks.

**Proposition 6.1** (RoPE Gradient Seed Coherence). *Under Assumption (A2), the aggregated anchor gradient for RoPE, $H_L^{(h)}(d)$, has a deterministic lower bound. A strictly positive seed ($H_L^{(h)}(d) > 0$) is guaranteed if the learning signal is sufficiently strong to satisfy the condition $a_*^{(h)}(d)\, \eta_*^{(h)}(d) > C^{(h)} \chi_L$.*

**Proposition 6.2** (ALiBi Gradient Cancellation). *Under Assumptions (A3), for any empirical distance distribution $\hat{\mu}_N$, there exists a batch of samples realizing $\hat{\mu}_N$ such that the aggregated anchor gradient for ALiBi vanishes for all heads and distances: $H_L^{(h)}(d) \equiv 0$.*

The existence of a non-cancellable seed in RoPE is the starting point. The following theorem shows how this seed inevitably leads to the deposit pattern.

**Theorem 6.1** (Exponential Amplification and Specialization). *Let $\mathcal{U}$ be the subspace of gradient directions corresponding to the positive RoPE seeds from Proposition 6.1. The signal-to-noise ratio of the gradient within this subspace is amplified exponentially during backpropagation. Crucially, the margin between the gradient signal for the dominant head and that of the second-dominant head also grows exponentially.*

$$Margin_l \geq Margin_L \prod_{k=l}^{L-1} \gamma_k, \quad \text{where each gain factor } \gamma_k > 1. \tag{5}$$

**Implication.** Theorem 6.1 provides the theoretical basis for the single-head deposit pattern. Even a minuscule initial advantage for one head in learning the position-sensitive task (a non-zero $Margin_L$) becomes exponentially magnified as gradients propagate to lower layers. This dynamic strongly incentivizes the optimizer to allocate the entire task of positional reasoning to that single head, as its parameters receive a disproportionately larger learning signal, thus explaining the observed specialization. The detailed proofs for these results are provided in Appendix C.

## 7 CONCLUSION AND LIMITATIONS

In this work, we introduced a novel framework that leverages spectral theory to deconstruct how positional encoding (PE) mechanisms operate within the Transformer attention matrix. We demonstrated that this framework provides a powerful lens, allowing us to predict learning dynamics and uncover previously hidden processing patterns. Our primary discovery, the "single-head deposit pattern" in RoPE, serves as a clear mechanistic explanation for the observed paradoxes in its performance. By identifying the distinct signatures of additive and multiplicative coupling, our analysis not only diagnoses the cause of this over-specialization but also points toward a solution. The success of a hybrid architecture validates our theory and suggests that controlled mixing of positional signals is a promising strategy for achieving both spectral stability and robust, distributed representations. Furthermore, we exploit gradient flow to verify that this is an intrinsic property of the RoPE.

**Limitations:** Firstly, the connection we hypothesize between the deposit pattern and RoPE's known struggles with length extrapolation requires direct empirical validation. Secondly, future work should probe the limits of these positional mechanisms on more complex, algorithmic tasks such as sequence reversal or Dyck language recognition. However, most of these tasks are beyond the capabilities of transformers, so more complex controlled analysis is a serious challenge for future work.

ACKNOWLEDGMENTS

This work was supported by the National Natural Science Foundation of China under Grant 62372448.

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

## ACKNOWLEDGMENT OF LLM USAGE

During the preparation of this manuscript, large language models (LLMs) were employed in a limited and auxiliary capacity. Specifically, their usage was restricted to the following three aspects: (1) checking grammar and expression at the sentence level, thereby providing local linguistic refinement; (2) performing global polishing after the draft was completed, ensuring that the overall exposition conforms to idiomatic English usage; and (3) improving the readability of the proof details presented in the appendix.

At no stage were LLMs used for generating research ideas, developing arguments, or modifying the substantive content of this work. Their sole role was to assist in enhancing clarity and effectiveness of communication.

## A  NOTATIONS LIST

The notations used throughout this article are summarized in Table 2.

Table 2: Some important notations used in this paper.

| Notation | Description |
|---|---|
| $x_i$ | Real vector in $\mathbb{R}^{2n}$ |
| $\mathbf{x}_i$ | Complex vector in $\mathbb{R}^n$ |
| $c$ | Content label for a token or a hidden activation |
| $p$ | Position label for a token or a hidden activation |
| $e_i$ | Absolute position encoding for $i$ |
| $b$ or $\mathbf{B}$ | The bias decided by the relative position encoding |
| $R_i$ | The rotary matrix for position $i$ of RoPE |
| $G_{u,v}$ | The gram matrix for the vector columns $\{v_i\}, \{u_i\}$ |
| $\mathbf{L}$ | Logit value of attention layer |
| $T_N(a)$ | Toeplitz Matrix for $a_i, i = -N, ..., N$ |

## B  DETAILED METHOD ANALYSIS

In this section of the appendix, we provide detailed theoretical analysis and derivations supporting the non-trivial conclusions presented in the main text. Our analysis relies on idealized proofs that simplify the complex optimization dynamics of neural networks, aiming to reveal a theoretical explanation for the observed experimental phenomena. While the actual simulation and analysis of non-convex optimization in full Transformer models is extremely difficult, this theoretical framework offers valuable insights into the underlying principles.

Our analysis is predicated on a core theoretical link regarding the role of spectral properties in learning. We rely on the established principle that favorable spectral properties of matrices involved in optimization—such as a more contracted eigenvalue spectrum (tighter bounds) or a smaller spectral norm—correlate with faster and more stable convergence of learning algorithms. Building upon this, our theoretical framework posits that the rapid and stable convergence of the positional coupling function, driven by these favorable spectral properties, will lead to a specific phenomenon during training: the localization of this function within a limited set of attention heads in shallow layers, resulting in the empirically observed deposit pattern Boyd & Vandenberghe (2004).

**Proposition B.1 (RoPE Logit Form):** This proposition formally derives the mathematical form of the Rotary Positional Encoding (RoPE) attention logit as presented in the main text (Equation 9), showing how RoPE's rotation mechanism translates into a specific inner product structure involving content and relative positional information. We would prove that

$$\mathbf{L}_{\text{RoPE\_complex}} = (G_{\mathbf{q,k}} + G_{\mathbf{q,p}} + G_{\mathbf{p,k}} + G_{\mathbf{p}}) \circ G_{\mathbf{e}}.$$

We use the fomula in Su et al. (2024) that a logit value of i and j in $\mathbf{L}_{\text{RoPE\_complex}}$ is

$$\sum_{k=0}^{d/2-1} \mathbf{q}_i^{(t)} \mathbf{k_j}^{(t)} e^{\mathrm{i}(i-j)\theta_t},$$

and there are $t_1$ and $t_2$ that

$$\big( \sum_{k=0}^{d/2-1} \mathbf{q}_i^{(t)} \mathbf{k_j}^{(t)} \big) e^{\mathrm{i}(i-j)\theta_{t_1}} \leq \sum_{k=0}^{d/2-1} \mathbf{q}_i^{(t)} \mathbf{k_j}^{(t)} e^{\mathrm{i}(i-j)\theta_t} \leq \big( \sum_{k=0}^{d/2-1} \mathbf{q}_i^{(t)} \mathbf{k_j}^{(t)} \big) e^{\mathrm{i}(i-j)\theta_{t_2}},$$

Due to continuity and the mean value theorem, there exists $t_{i-j}$ that

$$\sum_{k=0}^{d/2-1} \mathbf{q}_i^{(t)} \mathbf{k_j}^{(t)} e^{\mathrm{i}(i-j)\theta_t} = \big( \sum_{k=0}^{d/2-1} \mathbf{q}_i^{(t)} \mathbf{k_j}^{(t)} \big) e^{\mathrm{i}(i-j)\theta_{t_{i-j}}} = \mathbf{q_i} \mathbf{k_j} e^{\mathrm{i}(i-j)\theta_{t_{i-j}}}.$$

Then we can use $e^{\mathrm{i}(i-j)\theta_{t_{i-j}}}$ to generate $G_e$ to finish our proof.

**Proposition B.2 (Logit Expressiveness):** This proposition analyzes and demonstrates the higher degree of freedom and expressive power of the RoPE attention logit form (Equation 9) compared to the non-RoPE additive logit form (Equation 8). It highlights why RoPE's structure is theoretically better suited to capture the complex content-relative positional relationships required by tasks like Task 1.

Task 1 is directly related to the following optimization problem:

For $-i \leq s \leq d - s$ and constant $C_1$ and $C_2$,

1. **Relative Position Encoding**

$$< q_i, k_j > + b_{i-j} > C_1 > C_2 > < q_{i+s}, k_{j+s} > + b_{i-j},$$

2. **RoPE**

$$\mathbf{Re}\{< \mathbf{q}_i, \mathbf{k}_j > e^{\mathrm{i}(i-j)\theta_{t_{i-j}}}\} > C_1 > C_2 > \mathbf{Re}\{< \mathbf{q}_{i+s}, \mathbf{k}_{j+s} > e^{\mathrm{i}(i-j)\theta_{t_{i-j}}}\}.$$

It is easy to verify that the solution space of RoPE is much larger than that of Relative Position Encoding. This can be considered from the following two aspects. First, the expression of RoPE can make the key-value pairs with a fixed distance meet the requirements as long as the angle distribution meets the requirements, while the expression of Relative Position Encoding requires a more stringent angle distribution because the bias has no significant effect. Second, RoPE actually only needs a significant angle to easily meet the conditions, which actually corresponds to the cause of Massive Value described in Jin et al. (2025).

The emergence of the deposit pattern may share a common theoretical origin: the **spectral properties** of the attention logit matrix. Gradient-based optimization is known to be more stable for matrices with tightly clustered eigenvalues (Boyd & Vandenberghe, 2004). As suggested by Szegő's theorem (Grenander, 1958), the multiplicative interaction in RoPE is theorized to contract the eigenvalue spectrum of the positional component more effectively than additive methods.

**Proposition B.3 (Spectral Contraction):** This proposition formally proves the spectral property claimed in the main text. By applying theorems like Szegő's theorem and inequalities concerning the Hadamard product, it demonstrates that the Toeplitz matrix structure associated with multiplicative coupling (as in RoPE) has a more desirable eigenvalue spectrum (e.g., more compact or tighter bounds) compared to Toeplitz structures associated with additive coupling mechanisms.

The structure of the attention logit matrix $\mathbf{L}$, particularly the properties of its Toeplitz or Toeplitz-like components, provides crucial insights into the stability and dynamics of positional information processing. The width of the eigenvalue spectrum often has a great impact on the optimization process.

For Toeplitz matrices, classical results such as Szegő's theorem connect matrix structure to asymptotic eigenvalue distributions. We recall a relevant form below:

**Theorem B.1** (Szegő's Theorem on Eigenvalue Distribution (From Grenander (1958))). *Let $T_N(a)$ be an $N \times N$ Toeplitz matrix with $(T_N(a))_{i,j} = a_{i-j}$, and let $a(e^{i\theta}) = \sum_{k=-\infty}^{\infty} a_k e^{ik\theta}$ be its symbol. If $a(e^{i\theta})$ is real-valued and continuous, then the eigenvalues $\lambda_j^{(N)}$ of $T_N(a)$ asymptotically fill the interval $[\min a(e^{i\theta}), \max a(e^{i\theta})]$ as $N \to \infty$.*

Applying this theorem and related spectral analysis techniques to the Toeplitz structures induced by various PE mechanisms:

- In Non-RoPE methods, the Toeplitz components $G_{q^p,k^p}$ and $\mathbf{B}$ contribute additively to the logit, and their spectral ranges are determined by their respective coefficient sequences.
- In RoPE, the complex logit involves a Hadamard product with the Toeplitz matrix $G_\mathbf{e}$. This multiplicative interaction theoretically contracts the eigenvalue range relative to additive compositions, resulting in tighter spectral bounds.

We simplify the proposition into the following: Let $W, E$ be $N \times N$ positive definite Toeplitz matrices. Then the eigenvalue spectrum of the Hadamard product $W \circ E$ is more compact than $W$.

Here, each element of E is a complex number with a membrane length of 1. We can directly use Schur's inequality to prove that the membrane length of the eigenvalue of $W \circ E$ must be less than or equal to the membrane length of the eigenvalue of $W$.

However, a strict comparison requires citation of Szegő's Theorem B.1 on Eigenvalue Distribution Grenander (1958). So let us calculate the generating function (sign function) corresponding to $W \circ E$ and $W$ respectively.

$$\mathbf{Symble}(W \circ E)(\theta) = \sum_{i=-N+1}^{N-1} w_i e^{\mathrm{i}(i\theta_i + i\theta)} = \sum_{i=0}^{N-1} 2\mathbf{Re}\{w_i\} \cos(i\theta_i + i\theta)$$

$$\mathbf{Symble}(W)(\theta) = \sum_{i=-N+1}^{N-1} w_i e^{\mathrm{i}(i\theta)} = \sum_{i=0}^{N-1} 2\mathbf{Re}\{w_i\} cos(i\theta)$$

We will use the following lemma to prove that the range of the above formula must be smaller than that of the following formula, and thus Szegő's Theorem shows that its eigenvalue spectrum is more compact.

**Lemma B.1** (Amplitude bound and phase–alignment criterion). *Fix an integer $N \geq 1$ and non-negative weights $w_i \geq 0$. Define*

$$f(\theta) := \sum_{i=0}^{N-1} 2w_i \cos(i\theta_i + i\theta), \qquad g(\theta) := \sum_{i=0}^{N-1} 2w_i \cos(i\theta), \qquad \theta \in \mathbb{R}.$$

*Then*

$$\max_{\theta \in \mathbb{R}} |f(\theta)| \leq \max_{\theta \in \mathbb{R}} |g(\theta)| = 2 \sum_{i=0}^{N-1} w_i.$$

*Equality holds **iff** all phase offsets coincide modulo $2\pi$:*

$$\theta_0 \equiv \theta_1 \equiv \cdots \equiv \theta_{N-1} \pmod{2\pi}.$$

*Consequently the peak-to-peak range of $g$ (width $4 \sum_i w_i$) never falls below that of $f$, and both ranges coincide only under complete phase alignment.*

*Proof.* Denote

$$P(z) := \sum_{i=0}^{N-1} c_i z^i, \quad c_i := w_i e^{i\theta_i}, \qquad Q(z) := \sum_{i=0}^{N-1} w_i z^i, \quad |z| = 1.$$

**Upper bound.** For any $|z| = 1$ the triangle inequality gives

$$|P(z)| \leq \sum_{i=0}^{N-1} |c_i| = \sum_{i=0}^{N-1} w_i.$$

Hence

$$|f(\theta)| = 2|\Re P(e^{i\theta})| \leq 2|P(e^{i\theta})| \leq 2\sum_{i=0}^{N-1} w_i = |g(0)|.$$

Conversely, $|\cos(i\theta)| \leq 1$ implies $|g(\theta)| \leq 2\sum_i w_i$ for every $\theta$, while $g(0) = 2\sum_i w_i$ shows that this bound is attained, so $\max_\theta |g(\theta)| = 2\sum_i w_i$. Combining the two yields the desired inequality.

**Equality case.** Suppose $\max_\theta |f(\theta)| = 2\sum_i w_i$. Then at some $\theta^\star$ both previous inequalities are tight:

$$|P(e^{i\theta^\star})| = \sum_i w_i, \qquad |\Re P(e^{i\theta^\star})| = |P(e^{i\theta^\star})|.$$

The first equality forces **all terms in the sum for $P$ to share a common phase**, i.e. $e^{i(\theta_i + i\theta^\star)}$ is the same for every $i$; the second forces this common phase to be 0 or $\pi$, which does not affect alignment. Thus

$$\theta_i + i\theta^\star \equiv \theta_0 \pmod{2\pi} \quad \forall i \implies \theta_i \equiv \theta_0 \pmod{2\pi} \quad \forall i.$$

Conversely, if all $\theta_i$ are equal, choosing $\theta = -\theta_0$ gives $f(-\theta_0) = g(0) = 2\sum_i w_i$, so equality is achieved.

The lemma follows. $\qquad\square$

## C  DETAILED PROOFS FOR THEORETICAL ANALYSIS

This appendix provides the detailed proofs for the propositions and theorems presented in Section 6.

### C.1  PRELIMINARIES AND NOTATION

We consider a Transformer with $L$ layers and $H$ heads. For a batch of $N$ samples, $(i_s, j_s)$ are the token positions for sample $s$, and $d_s = |j_s - i_s|$ is the relative distance. The empirical measure of distances is $\hat{\mu}_N(d) := \frac{1}{N}\sum_{s=1}^{N} \mathbf{1}\{|j_s - i_s| = d\}$. The attention mechanism is defined by $S^{(h,l)} = Q^{(h,l)}K^{(h,l)\top}/\sqrt{d_h}$, $A^{(h,l)} = \text{softmax}_{\text{row}}(S^{(h,l)})$, and $Y^{(h,l)} = A^{(h,l)}V^{(h,l)}$. We use the following constant, derived from bounded operator norms and loss gradients:

$$C^{(h)} := \|W_O\|_2 \|V^{(h,L)}\|_2 \sup_s \|\partial\ell_s/\partial y\|_2. \tag{6}$$

We also denote the row sharpness as $\chi_L := \max_i \|A_{i,:}^{(h,L)}\|_2$ and the minimum anchor attention as $a_*^{(h)}(d) := \inf_{s:\, d_s = d} A_{u_s, j_s}^{(h,L)}$.

### C.2  SOFTMAX GRADIENT CALCULUS

**Lemma C.1** (Row-softmax Gradient Identity). *For any layer $l$, head $h$, sample $s$, and row $i$,*

$$\frac{\partial\ell_s}{\partial S_{i,j}^{(h,l)}} = A_{i,j}^{(h,l)}\Big(\Lambda_{i,j}^{(h,l)} - \langle\Lambda_{i,:}^{(h,l)}, A_{i,:}^{(h,l)}\rangle\Big), \tag{7}$$

*where $\Lambda^{(h,l)} := (\partial\ell_s/\partial Y^{(h,l)})V^{(h,l)\top}$.*

*Proof.* The result follows from the chain rule. The Jacobian of a row-wise softmax is $J_i = \text{diag}(A_{i,:}) - A_{i,:}A_{i,:}^\top$. The upstream gradient is $\partial\ell/\partial A = (\partial\ell/\partial Y)V^\top$. Combining these yields the identity. $\square$

**Lemma C.2** (Quantitative Anchor Gain). *Let $j^\star$ be an anchor column for row $i$. Let $J_i$ be the softmax Jacobian for row $i$. For the canonical basis vector $e_{j^\star}$, we have:*

$$\|P_{e_{j^\star}} J_i^\top e_{j^\star}\|_2 = A_{i,j^\star}(1 - A_{i,j^\star}) \quad and \quad \|P_{(e_{j^\star})^\perp} J_i^\top e_{j^\star}\|_2 \leq A_{i,j^\star}(1 - A_{i,j^\star}). \tag{8}$$

*Proof.* A direct computation gives $J_i^\top e_{j^\star} = A_{i,j^\star}e_{j^\star} - A_{i,j^\star}A_{i,:}$. The component along $e_{j^\star}$ is $A_{i,j^\star}(1 - A_{i,j^\star})$. The squared norm of the orthogonal component is $A_{i,j^\star}^2 \sum_{k \neq j^\star} A_{i,k}^2 \leq A_{i,j^\star}^2(\sum_{k \neq j^\star} A_{i,k})^2 = A_{i,j^\star}^2(1 - A_{i,j^\star})^2$. Taking square roots yields the result. $\square$

## C.3 PROOF OF PROPOSITION 6.1 (RoPE TOP-LAYER SEED)

*Proof.* From Lemma C.1, for a sample $s$ with anchor $u_s$ and target $j_s$:

$$\frac{\partial\ell_s}{\partial S_{u_s,j_s}^{(h,L)}} = A_{u_s,j_s}^{(h,L)}\left(\Lambda_{u_s,j_s}^{(h,L)} - \langle\Lambda_{u_s,:}^{(h,L)}, A_{u_s,:}^{(h,L)}\rangle\right). \tag{9}$$

By Assumption (A3), the first term $\Lambda_{u_s,j_s}^{(h,L)} = \langle\partial\ell_s/\partial Y_{u_s,:}^{(h,L)}, V_{j_s,:}^{(h,L)}\rangle \geq \eta_s^{(h)}$. For the second term, Cauchy-Schwarz and the definition of $C^{(h)}$ yield:

$$|\langle\Lambda_{u_s,:}, A_{u_s,:}\rangle| \leq \|\Lambda_{u_s,:}\|_2 \|A_{u_s,:}\|_2 \leq \|\partial\ell_s/\partial Y_{u_s,:}\|_2 \|V^{(h,L)}\|_2 \chi_L \leq C^{(h)}\chi_L. \tag{10}$$

Combining these, and using $A_{u_s,j_s}^{(h,L)} \leq 1$:

$$\frac{\partial\ell_s}{\partial S_{u_s,j_s}^{(h,L)}} \geq A_{u_s,j_s}^{(h,L)}\eta_s^{(h)} - A_{u_s,j_s}^{(h,L)}C^{(h)}\chi_L \geq a_*^{(h)}(d_s)\eta_*^{(h)}(d_s) - C^{(h)}\chi_L. \tag{11}$$

Summing over all samples $s$ with $d_s = d$ to get $H_L^{(h)}(d)$ gives the result:

$$H_L^{(h)}(d) \geq N\,\hat{\mu}_N(d)\left(a_*^{(h)}(d)\,\eta_*^{(h)}(d) - C^{(h)}\chi_L\right). \tag{12}$$

This is strictly positive if the learning signal term $a_*\eta_*$ outweighs the interference term $C\chi_L$. $\square$

## C.4 PROOF OF PROPOSITION 6.2 (ALiBi CANCELLATION)

*Proof.* We provide a constructive proof. Fix a distance bucket $d$ and partition its samples $\mathcal{S}_d$ into pairs $(s, s')$. For each pair, we construct their token embeddings to achieve cancellation.

1. **Identical Attention Matrix:** By (A3), choose $t$ such that $W_Q t = W_K t = 0$ and $W_V t \neq 0$. For a base token embedding $x_0$, set the anchor token embedding for sample $s$ to be $x_s = x_0 + t$ and for $s'$ to be $x_{s'} = x_0 - t$. All other token embeddings are identical for $s$ and $s'$. Since the query and key projections of $t$ are zero, all dot products $q_i \cdot k_j$ are identical for both samples. As ALiBi adds the same bias $b_h(j - i)$, the scores $S^{(h,L)}$ and attention matrices $A^{(h,L)}$ are identical for $s$ and $s'$.

2. **Flipped Gradient Signal:** Since $W_V t \neq 0$, the value vector at the anchor row $u_s$ is flipped: $V_{u_s,:}^{(h,L)}(s') \approx V_{u_s,:}^{(h,L)}(s) - 2W_V t$. By appropriate choice of $x_0$, this can be made an exact sign flip. This implies that the term $\Lambda_{u_s,:}^{(h,L)}(s') = -\Lambda_{u_s,:}^{(h,L)}(s)$.

3. **Pairwise Cancellation:** Applying Lemma C.1 to the anchor edge $(u_s, j_s)$ for both samples, we find their respective gradients are equal and opposite, summing to zero. Repeating this for all pairs makes $H_L^{(h)}(d) = 0$.

$\square$

## C.5 PROOF OF THEOREM 6.1 (EXPONENTIAL AMPLIFICATION)

Let $\mathcal{U}$ be the subspace spanned by gradient directions corresponding to the seed-positive buckets. Let $g^{(l)}$ be the gradient vector at layer $l$, so $g^{(l)} = J^{(l)\top} g^{(l+1)}$.

**Lemma C.3** (Layerwise Directional Advantage). *Under the assumptions, there exists $\beta_l \geq 0$ such that for any $v \neq 0$,*

$$\frac{\|P_{\mathcal{U}} J^{(l)\top} v\|_2}{\|P_{\mathcal{U}^\perp} J^{(l)\top} v\|_2} \geq \frac{1 - \alpha_l}{1 + \alpha_l}(1 + \beta_l). \tag{13}$$

*The gain $\beta_l$ is lower-bounded by a term proportional to the summed strength of the positive seeds from Proposition 6.1.*

*Proof.* The Jacobian $J^{(l)}$ is decomposed into the within-attention part $\mathcal{O}^{(l)}$ and the rest of the block $\mathcal{R}^{(l)}$. By (A1), the $\mathcal{R}^{(l)}$ part contributes the factor $(1 - \alpha_l)/(1 + \alpha_l)$. The gain $\beta_l$ arises from $\mathcal{O}^{(l)}$. Lemma C.2 establishes a directional gain for the anchor component within the softmax Jacobian. When composed with the value projection and the upstream gradient from (A2), the anchor direction receives a coherent signal. Aggregating over the anchor rows in subspace $\mathcal{U}$ yields a net directional advantage, which extends from the basis vectors of $\mathcal{U}$ to any vector $v$ by a convexity argument. $\square$

*Proof of Theorem 6.1.* Applying Lemma C.3 to the backpropagation recurrence $g^{(l)} = J^{(l)\top} g^{(l+1)}$:

$$\text{SNR}_l = \frac{\|P_{\mathcal{U}} J^{(l)\top} g^{(l+1)}\|_2}{\|P_{\mathcal{U}^\perp} J^{(l)\top} g^{(l+1)}\|_2} \geq \gamma_l \cdot \frac{\|P_{\mathcal{U}} g^{(l+1)}\|_2}{\|P_{\mathcal{U}^\perp} g^{(l+1)}\|_2} = \gamma_l \cdot \text{SNR}_{l+1}. \tag{14}$$

Iterating this inequality from layer $L - 1$ down to $l$ yields the exponential product. The same multiplicative logic applies to the vector components corresponding to the top two heads within the subspace $\mathcal{U}$, proving margin amplification. $\square$

## C.6 REMARK: CONCEPTUAL LINK TO CONVOLUTION KERNELS

It is helpful to conceptually frame the aggregated gradient $H_L^{(h)}(d)$ as a discrete convolution. If we define the empirical distribution of distances as a signal $\hat{\mu}_N$, and the expected gradient contribution for a given relative distance $\Delta$ as a "deposit kernel" $\kappa^{(h)}(\Delta)$, then the total aggregated gradient is their convolution: $H_L^{(h)} = N(\hat{\mu}_N * \kappa^{(h)})$.

From this perspective, our proof demonstrates a key difference in the structure of these implicit kernels:

- For **RoPE**, the kernel $\kappa^{(h)}(\Delta)$ has a trigonometric structure due to the rotational mechanism. This structure is analogous to a positive-definite kernel, which resists being driven to zero and ensures a positive "seed" is deposited.

- For **ALiBi**, the kernel $\kappa^{(h)}(\Delta)$ is merely affine (linear plus a constant). This simple structure allows for exact cancellation when convolved with a symmetric distance distribution, explaining why no seed is guaranteed.

This intuitive framing aligns with the rigorous proof and reinforces the conclusion that the deposit pattern is an inherent property of RoPE's multiplicative design.

# D SUPPORT EXPERIMENT

## D.1 EXPERIMENT SETUP

All experiments were performed on a single NVIDIA RTX 4090.

The experimental parameter settings of non-MLA are shown in Table 3, and the experimental parameter settings of MLA are shown in Table 4.

*Remark* D.1. Note that all our experiments used dropout, so the deposit pattern is not a phenomenon that can be improved by ordinary dropout.

Since Task 2 did not have a position effect, our subsequent experiments were all on Task 1.

| Setting | Value |
| --- | --- |
| Vocabulary size (`vocab_size`) | 574 |
| Model dimension ($d_{\text{model}}$) | 256 |
| Feed-forward dimension ($d_{\text{ff}}$) | 512 |
| Number of attention heads (`num_heads`) | 16 |
| Number of decoder layers (`num_layers`) | 6 |
| Dropout rate | 0.1 |
| Maximum sequence length (`max_len`) | 128 |
| Positional encoding types tested | `nope`, `absolute`, `alibi`, `relative`, `random`, `rope` |
| Number of epochs | 100 (Task 1);150 (Task 2) |
| Batch size | 512 or 1024 |
| Optimizer | AdamW (lr=1e-4, betas=(0.98, 0.9), weight_decay=1e-5) |
| Learning-rate scheduler | Cosine schedule with warm-up (6% of total steps) |
| Loss function | Cross-entropy |
| Train/Test split | 70% / 30% |
| Random seed | 0,42,70,113,130 |

Table 3: Hyperparameters and settings (PE).

| Setting | Value |
| --- | --- |
| Vocabulary size (`vocab_size`) | 574 |
| Model hidden dimension ($d_{\text{model}}$) | 256 |
| Feed-forward dimension ($d_{\text{ff}}$) | 512 |
| Number of logical MLA heads (`num_heads`) | 16 (8 in fact) |
| Compression dimension ($d_{\text{compress}}$) | 128 |
| Number of decoder layers (`num_layers`) | 6 |
| Dropout rate | 0.1 |
| Maximum sequence length (`max_len`) | 128 |
| Positional embedding | Rotary (RoPE) |
| Number of epochs | 100 (Task 1); 150 (Task 2) |
| Batch size | 1024 |
| Optimizer | AdamW (lr=1e-4, betas=(0.98, 0.9), weight_decay=1e-5) |
| Learning-rate scheduler | Cosine warm-up (6% of total steps) |
| Loss function | Cross-entropy |
| Train/Test split | 70% / 30% |
| Random seed | 0,42,70,113,130 |

Table 4: Hyperparameters and settings (MLA).

## D.2 COMPLETE VISUALIZATION OF LAYER-BY-LAYER HEAD ABLATION

For completeness, we provide the full layer-by-layer head ablation results for the RoPE models trained on **Task 1**. These figures expand upon the summary violin plots in Fig. 4, showing the *per-head* accuracy after zeroing out each attention head at every layer.

**How to read the plots.** Each subplot corresponds to one Transformer layer, and each point plots the test accuracy after ablating a single head. Large vertical drops indicate heads whose removal severely harms performance, revealing where positional reasoning is concentrated. Flat curves indicate heads that contribute minimally and whose ablation produces no measurable degradation.

**6-layer RoPE model.** Figure 9 shows the full ablation for the 6-layer model. The first layer contains a single dominant head whose ablation causes a large accuracy collapse, while all remaining heads exhibit near-flat profiles across layers. This fine-grained visualization matches the condensed "deposit pattern" presented in the main text.

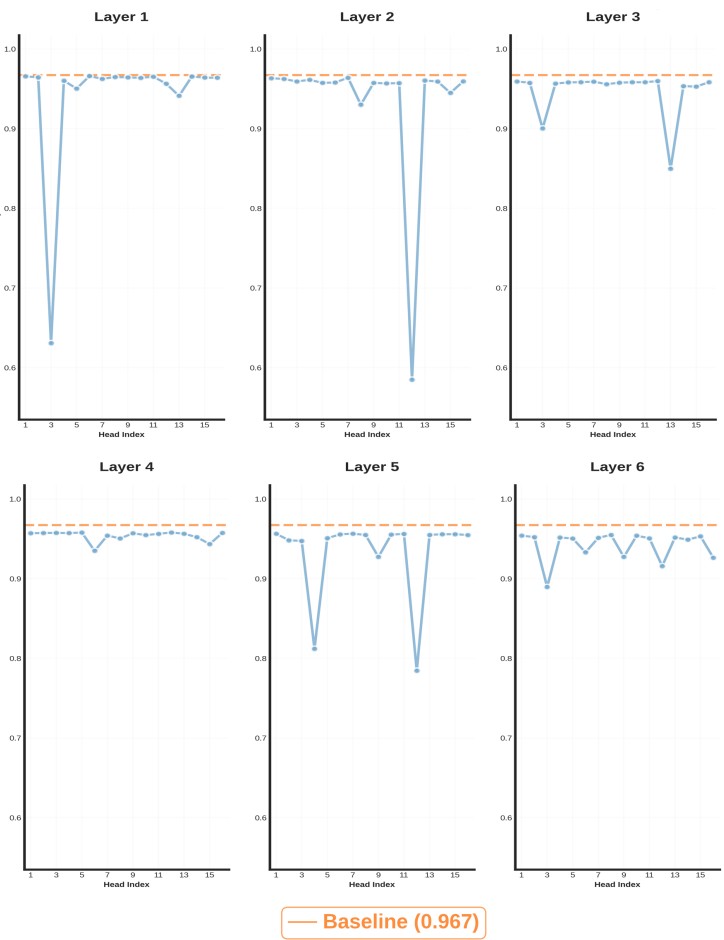

Figure 9: **Full head-ablation curves for the 6-layer RoPE model on Task 1.** Each subplot shows the test accuracy after ablating a single attention head in the corresponding layer. A single head in Layer 1 produces a catastrophic accuracy drop, while all other heads across all layers induce only minor changes. This confirms that RoPE concentrates positional reasoning into one early-layer head, even in shallower architectures.

**8-layer RoPE model.** Figure 10 reports the same analysis for an 8-layer RoPE model. Despite the increased depth, the pattern is qualitatively identical: a single early-layer head carries almost all positional reasoning, and deeper layers show only small, noise-level fluctuations across heads. This demonstrates that the deposit pattern is not sensitive to depth and remains stable across architectures.

**Summary.** These complete ablation maps confirm that the deposit pattern is a robust and highly localized effect: RoPE consistently routes positional reasoning into one head in the earliest layers, with all other heads acting effectively position-agnostic for this task.

## D.3 EXPERIMENTS OF HEAD ABLATION ON OTHER PE METHODS

In this subsection, we use violin plots to show the ablation experiment results of Alibi, Absolute PE, and Relative PE.

In Figure 11, we can find that the position information of relative position encoding and ALiBi are scattered, while the absolute position encoding has some significant heads and some insignificant heads in the shallow layer, and the whole is oscillating.

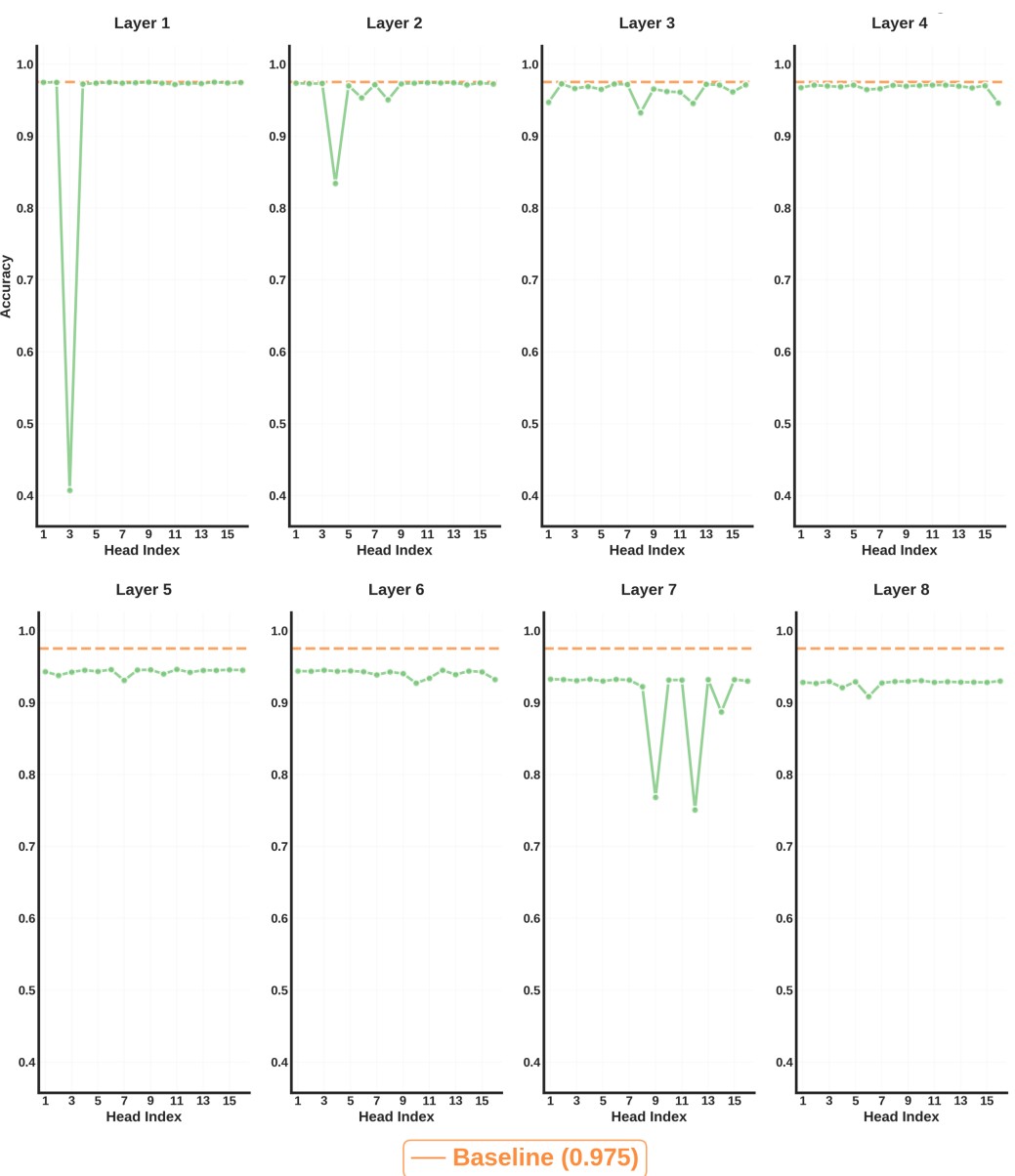

Figure 10: **Full head-ablation curves for the 8-layer RoPE model on Task 1.** The same localization pattern reappears at larger depth: a unique critical head in Layer 1 causes a large accuracy collapse, and all remaining heads across Layers 2–8 have negligible impact. The persistence of this topology across depth demonstrates the stability and robustness of the RoPE deposit pattern.

The similarities among NoPE, Relative PE, and ALiBi are consistent with previous studies on how NoPE obtains position information through causal masks Kazemnejad et al. (2023); Haviv et al. (2022).

### D.4    SOME EXPERIMENTS ON PARTIAL RoPE

It is foreseeable that MLA can fully alleviate the deposit pattern at the expense of a little generalization ability. We also conducted some ablation experiments in the experiments of RoPE for some heads and NoPE for some heads. Interestingly, we found that the most significant deposit pattern is not necessarily in the first layer, but may be in the second or third layer. In some cases, the first layer does not even have a deposit pattern. We can imagine that this is the effect of using NoPE attention

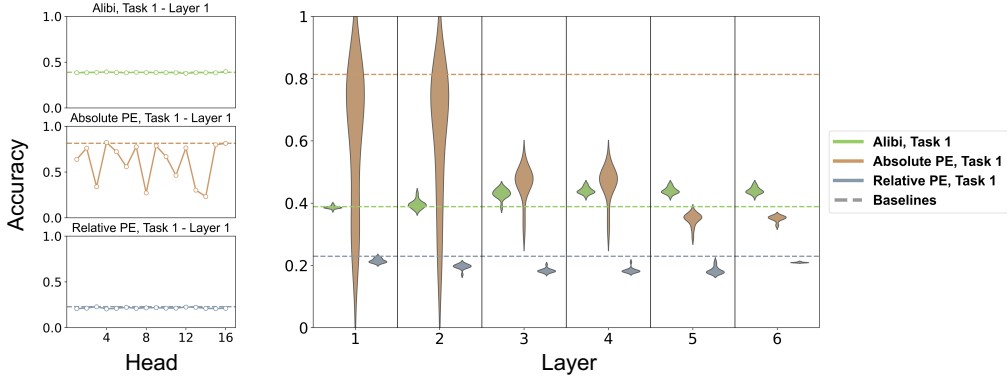

Figure 11: Ablation experiments on other PE methods.

heads in deeper layers. This inspires us that the way the FFN affects the position encoding is far more complicated than we imagined. We give the partial ablation line graph of the first k heads doing RoPE layer l below.

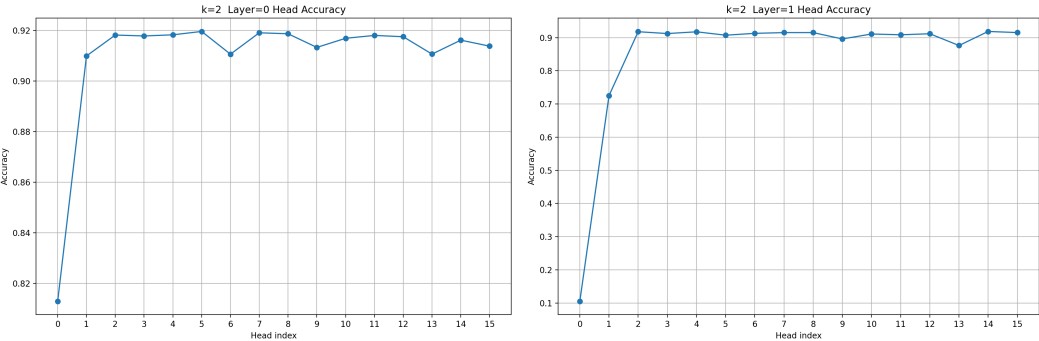

Figure 12: $k = 2, l = 1, 2$.

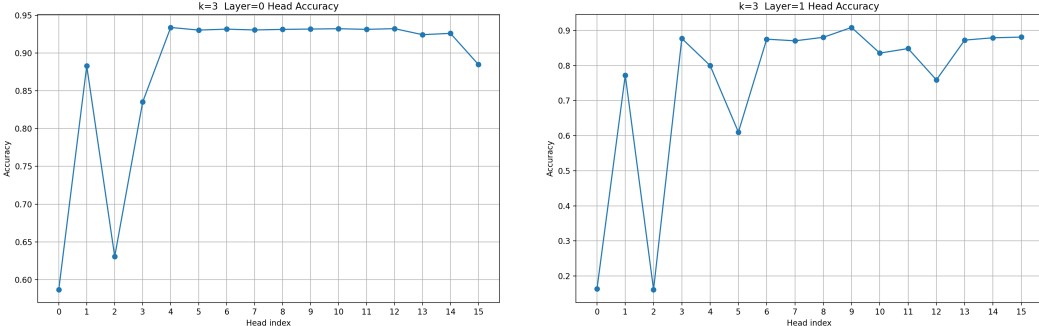

Figure 13: $k = 3, l = 1, 2$.

### D.5 WHY POSITION EMBEDDING COMPLEX?

We conclude this section by explaining why understanding positional encodings is difficult and almost impossible to decouple.

We first give our conclusion that the position component in the token comes from manifold embedding rather than subspace decomposition, which means that extracting the position component requires us

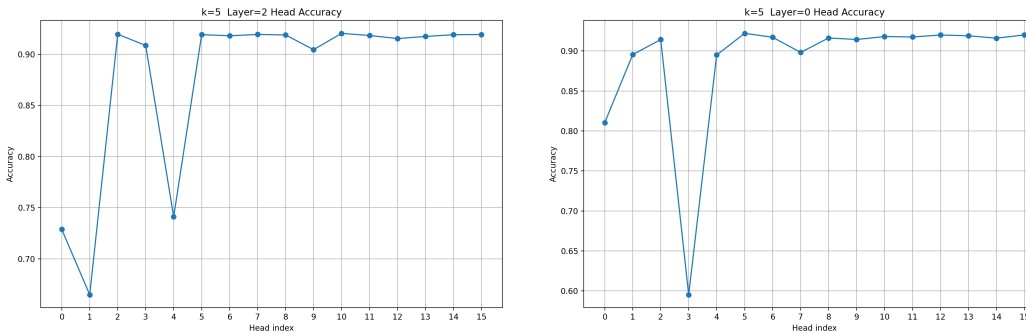

Figure 14: $k = 5, l = 1, 3$.

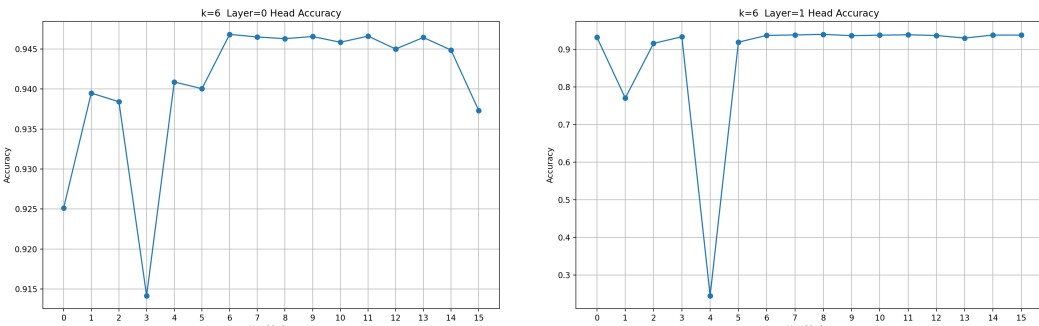

Figure 15: $k = 6, l = 1, 2$.

to have an algorithm to solve the changes in the position manifold. In an extremely ideal case, we can use the Levinson–Durbin algorithm in signal processing to recursively solve it through the value of the Toeplitz matrix. However, extracting a Toeplitz matrix from a general matrix is not unique, and we cannot determine whether the extracted Toeplitz matrix truly represents the position information.

This conclusion comes from a very simple observation: the position embedding of absolute position encoding is not orthogonal to word embedding. Therefore, there is no subspace decomposition that can completely separate the two components.

If we denote f as a continuous embedding from $S_1 \otimes S_1 ... \otimes S_1$ to $\mathbb{R}^n$, and g as a continuous embedding from $\mathbb{R}_+$ to $S_1 \otimes S_1 ... \otimes S_1$, then the essence of absolute position encoding is actually $f(g(i))$. In other words, the explicit position encoding that we can understand is essentially a manifold embedding.

The general method of dealing with manifold embeddings requires a specific embedding mapping. For general positional encodings, this mapping is not solvable, so it is almost impossible to decouple the positional encoding.

## E  RELATIONSHIP TO LENGTH GENERATION

Here, we give a more detailed qualitative statement to describe our thinking on the relationship between deposit patterns and length generation.

**Why the deposit pattern reflects a training bias rather than modularity.**    Although the deposit pattern superficially resembles a form of head-level modularity, our evidence suggests that it is instead a training-induced bias specific to RoPE's multiplicative structure. First, multiplicative PE suppresses activation-level positional directions (Section 5.4), causing early-layer heads to compete for the positional signal. Once a single head captures the content–position interaction, RoPE's strong inductive bias quickly eliminates positional gradients for the other heads, preventing them from

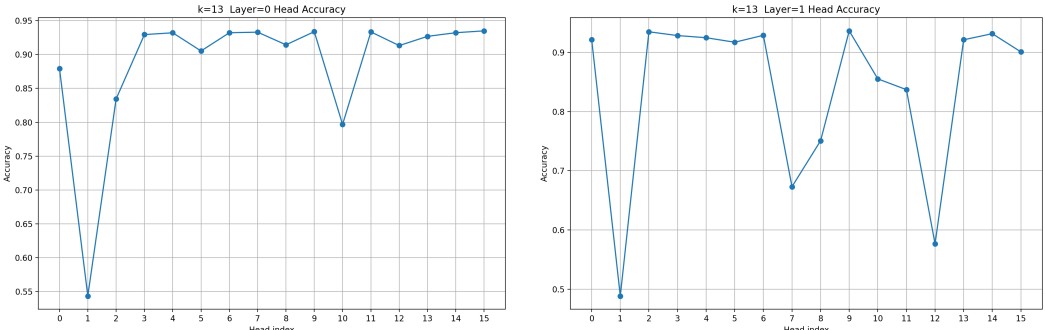

Figure 16: $k = 13, l = 1, 2$.

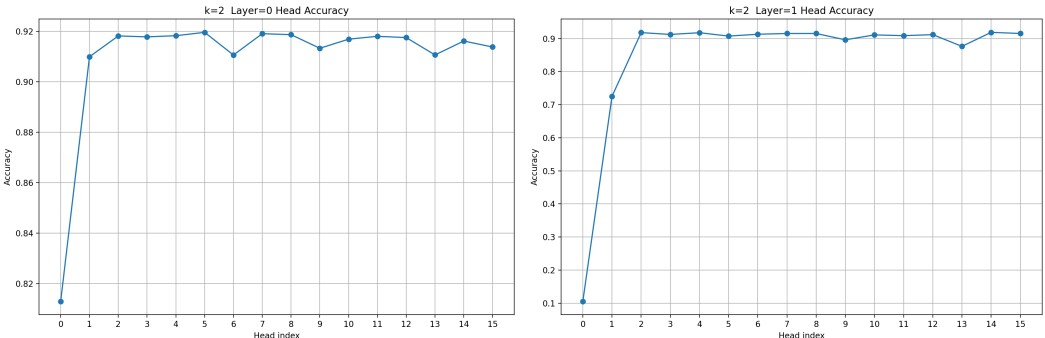

Figure 17: $k = 15, l = 2, 4$.

acquiring similar capabilities. This is fundamentally different from deliberate modularization: the specialization emerges not because the model decomposes the task across heads, but because the training dynamics collapse positional reasoning into the earliest head that happens to align with the RoPE kernel.

Second, the deposit head does not learn a general positional module; it learns a narrow set of frequency bands that are amplified by RoPE's rotation mechanism. As noted in Ermo et al. (2025), only a subset of frequencies is sufficiently trained, and their influence is later propagated through feed-forward mixing. This leads to a compressed, low-rank representation of positional structure, which is efficient for the training distribution but fragile outside it.

Taken together, the deposit pattern is better interpreted as a *training bias toward early collapse of positional responsibility*, rather than evidence for a stable or interpretable modular decomposition of positional reasoning. RoPE's multiplicative interaction creates a "winner-takes-all" dynamic, where one head monopolizes positional information not because the architecture encourages modularity, but because the optimization process and frequency structure favor such collapse.

**Further evidence from tasks beyond pairwise position reasoning.** RoPE enables efficient pairwise position–content coupling but does not equip the model with the compositional machinery needed for length-generalizing tasks that require combining several positional relations. To test whether the deposit head learned by RoPE represents a reusable "positional module," we designed a family of tasks requiring the model to combine multiple pairwise distances (e.g., predicting the sum of several trigger-word distances). Despite the task being a straightforward extension of our synthetic setup, all RoPE-based models failed to generalize beyond the training length, even when the training accuracy was near-perfect. In contrast to the pairwise distance task—where positional responsibility collapses into a single head—the multi-trigger task requires the model to compose several independent positional relations, a capability that is architecturally challenging for standard Transformers.

This observation aligns with prior findings on tasks such as Dyck languages, context-free composition, and stack-like dependencies (e.g., Chi et al. (2023)): Transformers struggle when a task requires retrieving, combining, or manipulating multiple positional relations simultaneously. The failure of RoPE to generalize in our multi-trigger setting therefore does not indicate a lack of training or capacity, but rather reveals that the single-head deposit behavior does *not* constitute a modular or compositional positional representation. Instead, the deposit head encodes a narrow, task-specific mapping tied to the training distribution, and—unlike a true module—it cannot be recombined or composed when the task demands multiple positional deductions.

Taken together, these results reinforce that the deposit pattern reflects a training-induced collapse of positional responsibility rather than the emergence of a reusable module. RoPE enables efficient pairwise position–content coupling but does not equip the model with the compositional machinery needed for length-generalizing tasks that require combining several positional relations.

More than one study has shown that transformer's understanding of position information starts from the first layer Zuo et al. (2025); Chi et al. (2023); Kazemnejad et al. (2023). The key to successful length generalization is that responses to relative position information within the training length can be naturally transferred to a longer length (test length). Therefore, the lower the coupling between location and content, the better the scalability. If the coupling degree is high, it is necessary to learn the coupling mode. At the same time, Deposit Patterns shows that only some frequency bands are fully trained during training like Ermo et al. (2025). These frequency bands are passed through the fully connected layer, spreading the influence to most of the frequency bands in the deep layer, which means that in fact only specific relative position differences are considered by the training process, so the length generalization ability is drastically affected.

**Relationship with massive value in RoPE.** Recent empirical studies have reported that Transformers trained with RoPE often develop a small number of extremely high-norm rows in $W_Q$ and $W_K$ (Jin et al., 2025), a behavior not observed under additive positional encodings. Although this phenomenon is not the focus of our work, the Toeplitz formulation naturally explains its underlying mechanism.

RoPE mixes content and position multiplicatively through a frequency-indexed Toeplitz kernel $G_{\mathbf{e}}$. In our formulation, each logit entry takes the form

$$\langle q_i, k_j \rangle_{\mathrm{RoPE}} = \langle q_i, k_j \rangle \circ (G_{\mathbf{e}})_{i-j},$$

where $(G_{\mathbf{e}})_{i-j}$ is a complex rotation encoding relative displacement. Importantly, each frequency band of RoPE corresponds to a fixed Toeplitz diagonal, and gradients flowing through that diagonal are accumulated entirely on the rows of $W_Q, W_K$ associated with the same frequency.

This creates a "frequency bottleneck":

- if a particular diagonal of the Toeplitz kernel becomes predictive early in training,
- then its associated frequency bands receive disproportionately large gradient flow,
- causing the corresponding rows of $W_Q, W_K$ to grow rapidly in norm,
- while other frequencies remain under-trained.

Thus the massive-value rows arise as a direct consequence of gradient concentration onto a small subset of Toeplitz diagonals, not from instability of the optimization process. Additive PEs do not exhibit this effect because their position signal enters only through additive $p_i$–based cross-terms, so no frequency-indexed diagonals receive concentrated multiplicative amplification.

Under our unified framework, the massive-value phenomenon is therefore a natural by-product of RoPE's multiplicative structure, in which positional information is stored directly in selected frequency bands of $W_Q, W_K$ rather than in explicit activation-level positional vectors.

