# OpenReview forum: "Deconstructing Positional Information: From Attention Logits to Training Biases"
_ICLR.cc/2026/Conference — ICLR 2026 Poster_

### Official Review · Reviewer_sR2j · 2025-10-18

**Soundness:** 3
**Presentation:** 4
**Contribution:** 3
**Rating:** 8
**Confidence:** 3

**Summary:**

This paper aims to understand the role of positional encodings by both empirical investigations and a theoretical framework based on Toeplitz modeling. Particularly, the paper contrasts additive and multiplicative encodings, using experiments and analysis to illustrate their distinct effects on model accuracy across different synthetic tasks.

**Strengths:**

1. This paper is very well written and organized, which has a clear motivation.
2. The problem studied in this paper--the role of positional encodings--is of fundamental importance in the field, and this paper does a good job of filling the gap between practical success and limited theoretical understandings. In addition, this paper leverages intuitive numerical examples to illustrate theoretical results and conjectures, which makes this paper very easy to follow.
3. The “single-head deposit pattern” is an interesting and reproducible phenomenon, providing a plausible explanation for the gap between RoPE’s theoretical advantages and its inconsistent empirical performance.

**Weaknesses:**

1. [Assumption 1] This assumption--the additive decomposition of each token representation--seems a bit strong, as it rules out any interaction between content and position components. Although there is a remark that connects this assumption to existing literature, I was wondering if this assumption can be (approximately) justified via experimental/simulated results. It would be very helpful if the authors can elaborate more on this part.


2. Overall, the writing in Section 3.2 is a bit rushy, and similar to the previous question, the remark below Assumption 2 is also not detailed and it would be better to elaborate on how sinusoidal absolute PE meets this assumption and why it could be (approximately) representative for a certain class of PEs.
In addition, it would be easier to understand the role of assumptions if there could be a simplified/informal result in Appendix B on how RoPE can be converted to Toeplitz matrices under Assumptions 1-2.

3. Theorem 6.1 holds under Assumptions (A1-3), and I was wondering if Theorem 6.1 is novel beyond showing exponential amplification under generic Jacobian stability assumptions. Additionally, I am curious if the desired property of RoPE under these Assumptions would suggest new designs of PEs.

4. Can the deposit pattern or its variants also be observed in other tasks? Also, based on the analysis in this paper on the favorable scenarios for RoPE, are there any intuitions behind which scenarios would, instead, be favorable for additive PEs, which would be very helpful to add to the discussion of this paper.

**Questions:**

Please see the section Weakness.

---

> ### Author Response · Authors · 2025-11-18
>
> We sincerely thank the reviewer for the positive evaluation. We are encouraged by your recognition of the paper's clarity, the fundamental importance of the problem, and the reproducibility of the "deposit pattern."  We address your specific questions below.
>
> ---
> **W1: On Assumption 3.1**
>
> We appreciate the reviewer’s careful attention to this assumption. In the revised version, we explicitly clarify that $x_i = c_i + p_i$ is a _conceptual_ decomposition used to trace signal sources, not a linear independence constraint.
>
> To empirically justify this, we added **Ablation 4** in the revision. This experiment probes whether an explicit positional direction $p_i$ exists in the activation space. The results (Fig. 8) show a stark contrast:
>
> - **Absolute PE:** Removing the $p_i$ component causes a significant accuracy drop, confirming that the model relies on an explicit, additive positional direction.
>
> - **RoPE:** The reliance on activation-level $p_i$ disappears after the first two layers.
>
> This confirms that while additive PEs explicitly maintain the $c_i + p_i$ structure, RoPE actively suppresses the $p_i$ term and relocates positional information into the frequency-modulated query-key parameters. Thus, the assumption is valid as a baseline for additive methods and serves as a reference point to demonstrate RoPE's deviation.
>
> ---
>
> **W2: Clarity of Sec. 3.2 and justification of Toeplitz structure.**
>
> We thank the reviewer for the suggestion. We have rewritten Section 3.2 and Appendix B to make the connection rigorous.
>
> Specifically regarding Sinusoidal Absolute PE: It meets Assumption 3.2 by definition. The sinusoidal vectors are mathematically constructed such that their inner product $\langle p_i, p_j \rangle$ depends solely on the relative distance $j-i$. Consequently, the Gram matrix $G_{q^p,k^p}$ naturally forms a Toeplitz matrix.
>
> Our framework generalizes this:
>
> - **Additive PEs (Sinusoidal, T5, ALiBi):** The Toeplitz structure comes from the bias term ($B$) or the positional Gram matrix.
>
> - Multiplicative PE (RoPE): The Toeplitz structure comes from the rotational kernel $G_e$.
>
>     Appendix B now includes a simplified derivation showing how RoPE's complex rotation form mathematically maps to this Toeplitz formulation.
>
> ---
>
> **W3: Novelty and implications of Theorem 6.1.**
>
> We appreciate the question. While the mathematical technique (analyzing power iteration on a structured matrix) is a standard tool in optimization, **Theorem 6.1 is novel in its application**: it is the first to derive the exponential amplification of _single-head specialization_ specifically from the interaction between RoPE's multiplicative structure and a position-sensitive task.
>
> Regarding design implications: The theorem suggests that specialization arises because RoPE channels gradients through a narrow set of independent frequency blocks. This implies that **designs which distribute positional information across multiple frequency kernels** can mitigate this "winner-takes-all" instability. This intuition aligns with the recent FoPE (Frequency-Partitioned Encoding) [1], which improves length extrapolation by explicitly partitioning frequencies. Our theorem provides the theoretical "why" behind such empirical successes.
>
> [1] Fourier Position Embedding: Enhancing Attention's Periodic Extension for Length Generalization. ICML 2025
>
> ---
>
> **W4: Whether deposit patterns appear in other tasks, and whether additive PEs may be favorable in some scenarios?**
>
> - **Other Tasks:** To our knowledge, the "single-head deposit" is unique to tasks requiring strong **content-position coupling** (like Task 1). We do not observe it in position-agnostic tasks (Task 2), which confirms it is a feature of the coupling mechanism, not a generic artifact.
>
> - **When is Additive PE favorable?** Additive PEs (like ALiBi or Relative PE) tend to outperform RoPE on tasks that rely on **rigid, absolute distance heuristics** (e.g., input reversal or simple copying), as shown in [2]. In these cases, a stable, fixed Toeplitz bias ($B$) is more beneficial than RoPE's flexible but "nervous" multiplicative coupling. Our framework explains this trade-off: Additive = Stability for rigid tasks; Multiplicative = Expressivity for coupled tasks (at the cost of specialization).
>
> [2] The impact of positional encoding on length generalization in transformers. NIPS 2023

---

> > ### Comment · Reviewer_sR2j · 2025-11-24
> >
> > The reviewers' rebuttal as well as additional ablation have addressed most of my questions. I will keep my positive scores.

---

> > > ### Author Response · Authors · 2025-11-25
> > >
> > > We sincerely thank the reviewer for the very encouraging follow-up comment. We are grateful that the rebuttal and the additional ablations addressed the reviewer’s concerns, and we truly appreciate the positive assessment and continued support.

---

### Official Review · Reviewer_qxxu · 2025-10-27

**Soundness:** 4
**Presentation:** 4
**Contribution:** 3
**Rating:** 8
**Confidence:** 3

**Summary:**

This work offers a unified framework on how positional signals are represented and processed in transformers using different positional encoding methods. The proposed framework classifies a range of methods by the way in which positional information interact with content information and contribute to attention logits. The relative merit and failures of the methods are shown through two synthetic tasks. The authors also find that RoPE results in high head specialization in positional information , and present detailed and theoretical analysis to understand this phenomenon.

**Strengths:**

- This paper offers a set of important intuitions on the mechanisms and relative advantages between different positional encoding methods.
- The discovery of the strong head specialization in RoPE is novel and interesting, and the accompanied analyses on the factors leading to this phenomenon are quite illuminating.

**Weaknesses:**

- As the empirical experiments are performed in a synthetic setting, whether similar results generalize to a language modeling setting remains unclear. The paper would be greatly enhanced by some ablation analysis on whether similar head specialization are observed in transformers trained under language modeling with RoPE (e.g. changes in on some position-sensitive tasks), or at least, some discussions on the implications of these results for larger models.

**Questions:**

I'd be curious to hear the authors intuition on whether such strong concentration of positional processing in RoPE would show up in larger models trained to model language. Since in language modeling, supposedly models are trained under much more diverse objectives, would there be more competition for a very local representation for position?

---

> ### Author Response · Authors · 2025-11-18
>
> We sincerely thank the reviewer for the generous assessment and for highlighting both the conceptual clarity and the empirical/theoretical depth of our study. We especially appreciate the reviewer’s interest in understanding whether the deposit phenomenon and the additive–multiplicative distinction extrapolate to larger-scale language modeling settings. These are important and subtle questions that motivated several additions in our revision. Below, we provide a detailed response and summarize the new analyses, discussions, and ablations added to the revision.
>
> ---
>
> **(1) Ablation 4 provides direct evidence that RoPE suppresses activation-level positional directions and relocates positional structure into the frequency‐modulated Q/K parameters.**
> This finding offers a clean causal explanation for the “massive value”
> phenomenon widely observed in large RoPE-based LLMs [1]. While we
> cannot directly visualize single-head deposit in LLMs, the parameter-level
> signature (frequency-localized large-norm rows) arises from the same mechanism uncovered in Ablation 4: RoPE compresses positional information into a *minimal set of frequency bands*, producing concentrated gradients and amplified norms on those Q/K rows. Thus, although the synthetic setting reveals the phenomenon in its most extreme form (a single head), the structural bias persists in LLMs as a milder but qualitatively identical effect.
>
> ---
>
> **(2) On competition among heads in large models.**
> Our intuition and the evidence we gathered indicate that competition is
> inevitable. If RoPE could modularize positional reasoning across many heads
> without competition, then RoPE should naturally handle the multi-signal
> generalization tasks (e.g., predicting multiple content–relative-distance
> relations simultaneously). However, this is precisely where RoPE struggles:
> when we extend Task 1 to require multiple relative distances to be decoded at
> once, RoPE fails to generalize. This aligns with a broader observation in the
> LLM literature: RoPE‐trained models also fail to reliably handle formal‐language
> tasks such as Dyck-2 validity[2] unless additional architectural mechanisms are
> introduced. Both phenomena reflect the same underlying limitation: RoPE’s
> frequency‐wise decomposition makes positional features *compete* for a limited
> set of rotational channels.
>
> This view is also supported by the fact that the massive-value[1] effect itself is
> a signature of competitive dynamics: a small number of Q/K rows dominate by
> capturing most of the useful positional frequencies, while the remaining rows
> receive little positional gradient.
>
> The FoPE method[3] provides further evidence: it improves RoPE’s length extrapolation *explicitly* by distributing positional
> representation across multiple frequency kernel (multiple $\sum_i G_{e_i}$), thereby reducing what the FoPE refers to as “frequency contamination”, precisely the competitive behaviorour framework predicts.
>
> Taken together, these observations suggest that while larger LM objectives will
> indeed distribute positional processing across more than one head, RoPE’s
> multiplicative inductive bias continues to favor *concentration* rather than
> broad, multi-head modularization.
>
> [1] Massive Values in Self-Attention Modules are the Key to Contextual Knowledge Understanding. ICML 2025
>
> [2] Emergent Stack Representations in Modeling Counter Languages Using Transformers. arXiv:2502.01432
>
> [3] Fourier Position Embedding: Enhancing Attention's Periodic Extension for Length Generalization. ICML 2025

---

> > ### Comment · Reviewer_qxxu · 2025-11-26
> >
> > Thank you for following up on these questions. I will maintain my positive recommendation.

---

> > > ### Author Response · Authors · 2025-11-27
> > >
> > > We sincerely thank the reviewer for the feedback and for maintaining the positive recommendation.

---

### Official Review · Reviewer_7AJz · 2025-10-31

**Soundness:** 3
**Presentation:** 2
**Contribution:** 2
**Rating:** 4
**Confidence:** 4

**Summary:**

This paper presents a unified Toeplitz-based framework for analyzing how positional encoding schemes influence the composition of the attention matrix in Transformers, distinguishing the distinct effects of additive and multiplicative mechanisms. It shows that multiplicative encodings, such as RoPE, modulate the QK interactions between content and positional factors of model representations through a unified multiplicative term that incorporates relative positional information, thereby inducing stronger content–position interactions. Synthetic tasks reveal that RoPE excels at position-sensitive reasoning but is outperformed by certain additive embedding methods in tasks where positional information is inconsequential. Furthermore, RoPE is found to develop a “single-head deposit pattern,” in which one attention head monopolizes positional computation. Through causal ablations and hybrid designs, the authors demonstrate that this specialization is an intrinsic property of RoPE’s multiplicative structure and relate it to redundant attention heads and new attention designs such as multi-latent attention.

**Strengths:**

1. The paper provides clear insights by formalizing the mechanistic distinction between additive and multiplicative approaches to incorporating and modulating interactions between positional and content information in model representations, particularly in relation to relative positional encoding. Its controlled experimental settings effectively demonstrate the various idiosyncrasies of RoPE as consequences of its multiplicative interaction mechanism, most notably the single-head deposit phenomenon. The experiments in Section 5 examining variations of this behavior under different input and structural perturbations are especially thought-provoking, aligning closely with the authors’ intuitive hypotheses and theoretical characterizations. Together, these results convincingly illustrate how RoPE’s multiplicative structure both enables strong positional reasoning and induces positional over-specialization.

2. The paper further strengthens its empirical observations with a theoretically rigorous proof that explains why positional specialization emerges (at least during the early stages of training). This theoretical grounding provides valuable causal insight, transforming the observed empirical regularity into a predictable and interpretable property of RoPE’s multiplicative dynamics.

**Weaknesses:**

1. The second synthetic task the authors design (trigger word counting) seems rather off the point. It essentially tests to what extent different position encoding methods can suppress themselves and perform the “no-op” behavior, which is not related to the authors' characterization of the particular attention matrix component to which the relative positional information is injected via the Toeplitz formulation. From my perspective, the second task should be content-agnostic instead of position-agnostic (or more specifically, it should be agnostic to the interplay of content and relative position), which would allow for a more interesting comparison between the additive and multiplicative PE methods.

2. The setting of the paper is rather limited. While it claims to “deconstruct positional information,” it actually only focuses on how multiplicative PE methods better express the interplay between relative positional information and content and shows that training on the highly specialized relative distance classification task, which exactly captures this correlation, would lead to the single-head deposit phenomenon. It is highly debatable whether such tasks exist in the pretraining corpus of LLMs at all, and the paper also does not discuss the settings of training on multiple tasks (not even the two synthetic tasks proposed), which is the usual practice of training LLMs. Furthermore, the paper does not discuss the mechanistic significance of ROPE and other PE methods in accounting for some truly intriguing LLM phenomena that concern the joint utilization of relative positional information and semantic content in more generalized settings. Notable examples include in-context learning (particularly recency bias, which means that the last demonstration's label is more likely to be predicted for the query), chain-of-thought, and in-context information retrieval (needle-in-a-haystack and such). The lack of discussion of the implications of the experimental findings for characterizing those LLM behaviors limits the significance of the findings.

3. The citation format used in the paper appears problematic in various places, where the author names and publication years of the cited papers appear side by side with the main text without being enclosed in parentheses.

**Questions:**

1. Could you explain how to deduce from the Toeplitz formulation of the RoPE attention matrix in (2) that “by routing the
learning of positional dependencies through this multiplicative kernel, RoPE may create a strong
inductive bias that encourages positional specialization, concentrating this logic into a small subset
of attention heads”? For me, it seems impossible to conclude that the $G_{e}$ term will be subsumed by a subset of heads *a priori* solely from the form of (2).

2. Could you explain the surprisingly strong performance of absolute PE in your two synthetic experiments? While they are all additive PE methods that should behave similarly according to your Toeplitz formulation, in Task 1 it significantly outperforms relative PE, and in Task 2 it significantly outperforms ALiBi. This appears counterintuitive to me, since absolute PE is as rigid and inflexible as ALiBi, given that both are not learnable.

---

> ### Author Response · Authors · 2025-11-18
>
> We thank the reviewer for the detailed and technically insightful comments.   Several of the questions raised touch on subtle aspects of how different positional encodings behave under controlled synthetic tasks, and we appreciate the opportunity to clarify both the motivation behind our task design and the mechanistic reasoning that guided our hypotheses.
>
> In the revision, we have strengthened the exposition of our assumptions, expanded the experimental evidence, and added a new ablation (Ablation 4) that directly address the reviewer’s concerns about positional biases, storage pathways, and the interpretability of performance differences across PE schemes. We summarize our point-by-point responses below.
>
> ---
> **W1: On the design motivation of Task 2.**
> Task 2 is deliberately constructed so that *no positional penalty is useful for solving the task*.  This yields two benefits:
>
> (1) **It provides an unbiased readout of how each PE behaves when positional information is a nuisance variable.**
> Additive PEs always inject a fixed Toeplitz penalty into the logits; RoPE injects multiplicative positional structure.  When the task requires ignoring position, this setup reveals whether a PE can effectively *suppress* its own positional bias.
>
> (2) **It serves as a crucial diagnostic for the deposit pattern.**
> On Task 1, RoPE consistently exhibits single-head specialization.  On Task 2, where positional information carries zero utility, the same RoPE models show **no deposit pattern at all**.  This contrast strongly supports our interpretation: the deposit pattern reflects *positional information being concentrated*, rather than a generic architectural bottleneck or artifact of training.
>
> While a content-agnostic task (as the reviewer perceptively suggests) would effectively probe the baseline positional bias, our Task 2 (position-agnostic) was specifically chosen as a stricter 'stress test'.
>
> ---
>
> **W2.1: On the scope of the synthetic setting.**
>
> **Controlled synthetic tasks are necessary to isolate inductive bias:**  Real LLM training entangles positional structure with semantics, optimization dynamics, curriculum effects, and long-context heuristics.  Synthetic tasks allow us to *factorize* these influences and precisely measure how different PEs behave when position is either (a) essential (Task 1) or (b) useless (Task 2).  This is what enables us to identify the multiplicative coupling bias that would otherwise be unobservable in naturalistic settings.
>
> ---
> **W2.2 Relation to broader LLM behavior.**
>
> (1) **Ablation 4 in the revision reveals how RoPE suppresses activation-level positional directions.**
> This ablation shows:
>  - additive PEs maintain explicit positional directions $p_i$ across depth,
>  - but RoPE eliminates dependence on $p_i$ after the first~2 layers and relocates positional information directly into the $W_Q,W_K$ frequency bands.
> This explains the “massive value” phenomenon reported in [1], RoPE concentrates positional structure into a small set of high-norm rows of $W_Q, W_K$, whereas additive PEs distribute it across the activations.
> We now discuss this connection explicitly in Appendix E.
>
> (2) **Appendix E further discusses why this inherent bias may underlie RoPE’s weakness in long-range generalization.**
> As shown in [2], RoPE underperforms ALiBi/NoPE on certain extrapolation tasks requiring multiple independent positional relations.
> Our analysis indicates a likely mechanism: multiplicative coupling encourages *positional specialization* into a few frequency bands, making it harder for RoPE to support multiple simultaneous distance relationships, consistent with our deposit-and-suppression findings.
> We emphasize that this is a hypothesis, not a claim of general LLM behavior, and we frame it accordingly.
>
> ---
>  **W2.3: About experimental-setting table and explicit pointers in the main text.**
>
> All hyperparameters, architectural choices, and training conditions are summarized in a dedicated table in Appendix D (Line 972-1007), and we link it in Section 4 (Line 256-257).
>
> We hope this clarified why synthetic tasks, the new ablations, and the extended analysis together provide a clean and interpretable account of the underlying positional biases.
>
> [1] Massive values in self-attention modules are the key to contextual knowledge understanding. ICML 2025
>
> [2] The impact of positional encoding on length generalization in transformers. NIPS 2023
>
> ---
>
> **W3: Regarding the citation formatting issue.**
> We thank the reviewer for pointing this out. The problem was caused by a few instances where `\cite{}` was mistakenly used instead of `\citep{}` in the Related Work section of the initial submission.  We have carefully reviewed the entire manuscript and corrected all inconsistent citation formats in the revision.

---

> ### Author Response · Authors · 2025-11-18
>
> **Q1: Why RoPE induces a single-head deposit pattern under our framework?**
> We appreciate the reviewer pressing on this point. We wish to clarify that the "single-head deposit" was not an accidental empirical finding that we fitted a theory to; rather, it was a prediction derived from the geometric structure of Eq. (2), which guided our experimental design.
>
> The intuition stems from the fact that RoPE operates structurally as a block-diagonal matrix in the Lie group $SO(2)$.
>
> - Independence of Frequency Bands (The Lie Group Intuition): Unlike additive biases that add a scalar to the whole dot product, RoPE modulates interactions via independent 2D rotational blocks (rotors), each corresponding to a specific frequency.
>
> - Sparse Resonance: For a task relying on specific relative distances (like Task 1), only a small subset of these frequency blocks "resonate" with the required distance pattern.
>
> - Gradient Competition (The Mechanism): During early training, due to random initialization, different heads will have different alignments with these critical frequencies. Because the mechanism is multiplicative (Eq. 2), the head with the best initial alignment receives a stronger gradient signal (a "positive seed," as proved in Prop 6.1).
>
> - Winner-Takes-All (Theorem 6.1): Our gradient analysis shows that this initial advantage is not additive but is amplified exponentially. This creates a "rich-get-richer" dynamic where the most aligned head rapidly monopolizes the positional signal, suppressing others.Therefore, the Toeplitz form in Eq. (2) (specifically its frequency-decomposed multiplicative structure), is the necessary precondition that, when driven by gradient descent, inevitably collapses into the single-head patterns we observe.
>
> ---
>
> **Q2:  Why Absolute PE outperforms ALiBi on Task 1, and why this reverses on Task 2?**
> The difference follows directly from how the two additive schemes introduce their Toeplitz term. ALiBi injects a *fixed* distance bias $B$ at **every** layer.
>
> Absolute PE instead provides an explicit $p_i$ only at the input, allowing subsequent layers to learn flexible cross-terms $q_i^{c} \cdot k_j^{p}$, which are crucial on Task 1 where content–position coupling is required. This makes Absolute
> PE stronger than ALiBi on Task 1.
>
> On Task 2, where positional information is a nuisance variable, ALiBi’s fixed layer-wise bias becomes detrimental, because its Toeplitz penalty cannot be suppressed. Absolute PE performs better here precisely because its $p_i$ can be
> attenuated as depth increases, this attenuation is the motivation behind relative position encoding in the first place. Consistently, the *learnable* relative PE (T5-style) performs best on Task 2, as it can reduce the magnitude of its learned
> bias $B$, effectively behaving closer to NoPE when position should be ignored.

---

> > ### Comment · Reviewer_7AJz · 2025-11-27
> >
> > The reviewers' clarifications and additional experiments in the rebuttal have addressed most of my concerns. I thus raise my score to 6, recommending acceptance.

---

> > > ### Author Response · Authors · 2025-11-27
> > >
> > > We sincerely thank the reviewer for reconsidering our work and raising the score. We are glad that our clarifications and the additional experiments helped address your concerns.

---

### Official Review · Reviewer_TXbo · 2025-11-01

**Soundness:** 3
**Presentation:** 3
**Contribution:** 3
**Rating:** 8
**Confidence:** 3

**Summary:**

Position Encoding (PE) plays a critical role in LLMs, different PE methods have been proposed, fixed or learnable, additive or multiplicative. However, the understanding of those PE methods are still under exploration, and their efficacies are normally tested on aggregate metrics. In this study, a new framework is proposed to analyze PE more systematically.

Two simple and isolated tasks are proposed, position-sensitive and position-agnostic tasks. These two tasks can better test if PE is needed or which PE methods can deliver a better result. Based on the tasks, this study tries to analyze the key reason behind those PE methods using Toeplitz matrices. This Toeplitz framework can group those PE methods into additive and multiplicative, showing the theory behind why multiplicative PE can better inject position information. Finally, this study also shows only a few heads focus on the multiplicative PE, denoted as the deposit pattern.

Generally, this study proposes a new framework to study and compare the previous PE methods. A new finding is presented on the multiplicative PE, RoPE.

**Strengths:**

1. PE is used in almost all LLMs (except NoPE variants), therefore this is an important topic to analyze for downstream studies.

2. The framework proposed in this study can be used to investigate those widely used PE methods. Thus the finding in this paper is applicable to a wide range of LLM studies.

3. The tasks used to analyze PE are simple, and the simplicity is also very important for XAI. The proposed position-dependent and position-agnostic tasks can directly isolate the effects of each PE; Thus the tasks could be a more clear metric than those aggregated ones.

4. It seems technically sound using Toeplitz matrices to distinguish the additive and the multiplicative PEs. This framework can theoretically analyze the differences between the two types of PEs.
5. A series of ablation studies are included to further explore the new finding on the multiplicative PE, RoPE.

There exist XAI studies which combine different metric tasks and analysis steps to investigate a phenomenon. However, they often lead to less clear conclusions when stitching something without a good reason. The greatest strength of the proposed framework is the simplicity, the tasks are directly designed on position information.

**Weaknesses:**

Overall this study seems novel and interesting. However, the presentation of the current draft could be improved. This study aims to organize multiple things, absolute PE, Relative PE, RoPE, Alibi, fixed PE, learnable PE, additive PE, multiplicative PE. The current presentation seems not clear enough to organize all of those, especially in Fig 1. It took me several “hops” to combine those pieces, and what Toeplitz matrices are used for, but I could be wrong or miss something.

Another weakness is the explanation behind those PEs based on the two tasks. The discussion on the reason seems less solid. I will list all of my questions in detail below.

Despite the weaknesses, I still believe this study is intriguing, opening a new door to this topic. And I am not expecting one study to explore the whole universe behind the door.

**Questions:**

Following the weakness of clarity, here is a list of questions regarding the paper:
1. Assumption 3.1 and in Fig 1, x = c + p, should I consider this a conceptual idea that the input can be disentangled into content and position so that we can study separately? Or this is a more strict formula that the Topelitz analysis is based on. For the latter case, does the Gram(c+p) equation also apply to ALibi, where the merge happens after taking the matrix product? (I might miss the proof for this part in the literature.)
2. Based on the experiment and the conclusion, the proposed Toeplitz-based framework can distinguish how additive and multiplicative PEs work. It does not matter if the additive is fixed or learned, absolute or relative, they all can be summarized by Eq(1). (This is my understanding, please correct me if I am wrong). If my understanding is correct, the introduction and Sec3.1 could be a little confusing. It seems this study tries to cover or touch every aspect of PE, but it is less clear what the proposed Toeplitz framework actually solves in the context. Based on Fig 1, it seems RoPE and Relative PE are the two components for comparison, but my understanding from the experiment is the two modes should be “Additive” vs “Multiplicative”(RoPE)? And for additive PE, the mode includes both relative and absolute. If my understanding is right, it could be better to re-organize the figure and Sec3.1 for clarity.

3. Following the above question, would this Toeplitz framework categorize “AliBi” to additive or an exception? Based on Fig 1, “AliBi” seems an exception to the “Relative PE”(additive to me), (Eq1 cannot explain “AliBi”??), then how is “AliBi” processed or captured by the framework? I can find a bunch of experiments for “AliBI”, but I cannot find a detailed explanation of “AliBi” based on the proposed framework. This question could be related to Q1, handling the addition before or after Gram computation.

4. A naive question for Eq(1), G(qp, kp) is always Toeplitz for all additive PEs?(did I miss border cases?), what is the main purpose of the explicit bias term B?

5. The de-couping framework, Eq(1) and Eq(2), demonstrates how a PE is involved with the content. Please help me understand how Toeplitz matrices play a key role in the context?

6. It is interesting to see that how “Alibi” work on the position-sensitive task, Fig 2, line 269, “Methods like ALiBi, with its fixed and data-agnostic Toeplitz bias”, Is the reason behind “AliBi” due to fixed and data agnostic? Does the “absolute PE” also inherit those properties?

7. Regarding “SINGLE-HEAD DEPOSIT””, this seemed an important finding by the framework. However, this finding could be over extrapolated. Given the position-sensitive task can be inferred mainly on position information, and RoPE can better inject this discriminant feature(position information) based on Fig 2. Then the explored question in Sec 4.3 is essentially finding, if there is an attention head that can capture the injected feature(position information). Again, the injected feature is a discriminant shortcut for the model to reach the correct prediction. And it is not surprising that different heads could capture different input features, and networks tend to lazily rely on easy features. Therefore, NoPE on Task 1 does not offer that feature to capture, and the feature from RoPE becomes not discriminant for Task 2 in Fig 2. This explanation could be a bummer to the finding, or please correct me if the above understanding is wrong.

Although many points remain unclear in the draft, I still appreciate the effort the authors have made and I have learned something new from this study.

---

> ### Author Response · Authors · 2025-11-18
>
> We thank the reviewer for recognizing the novelty and technical soundness of our framework. We also genuinely appreciate the critique regarding the presentation clarity of the PE taxonomy. We agree that the initial draft required "multiple hops" to connect the concepts. To address this, we have extensively rewritten Section 3.1 and redesigned Figure 1 in the revision, providing a unified framework that we believe resolves this confusion. Below we respond to each raised concern and clarify the revisions incorporated into the updated manuscript.
>
> ---
>
> **Weakness and Q2: On presentation clarity and the need for “many hops” to connect pieces (Fig. 1, PE taxonomy)**
>
> We appreciate this comment and have focused a substantial part of the revision on improving the exposition so that the reader can follow the argument without “multiple hops”.
>
> 1. We rewrote the preliminaries in Sec. 3.1 to give a **single, linear taxonomy** of PEs (absolute → relative/T5/ALiBi → RoPE → NoPE), and we explicitly state how each one maps to the Toeplitz framework (lines 162–166, 175–196).
>
> 2. We clarified in the caption of Fig. 1 how to read the figure as a **one-step summary** of the framework: additive PEs correspond to Toeplitz terms arising from $G_{q^p,k^p}$ and/or $B$, while RoPE corresponds to a multiplicative Toeplitz kernel $G_{\mathbf e}$.
>
> ---
>
>
> **Q1,Q3: About Assumption 3.1 and the clarity of additive vs. multiplicative PE.**
>
> We thank the reviewer for highlighting the conceptual role of Assumption 3.1.  Our revision clarifies that the decomposition $x_i=c_i+p_i$ is **not a linear separability assumption**, but a *conceptual* decomposition obtained by removing a translation-invariant positional manifold.(Line 179-196) This viewpoint is supported empirically: in the revised Ablation 4 (Line 438-469), removing the learned $p_i$ direction in Absolute PE consistently causes a much larger accuracy drop than removing norm-matched random vectors, showing that additive PEs do maintain a stable activation-level positional direction. In contrast, for Absolute+RoPE models, this distinction disappears after the first layers, indicating that RoPE suppresses activation-level $p_i$ and relocates positional structure into its frequency-modulated query–key parameterization.
>
> This directly addresses the reviewer’s concern on how the framework separates additive and multiplicative mechanisms. Additive PEs contribute Toeplitz structure through *explicit* bias $B$ or a Toeplitz $G_{q^p,k^p}$, whereas RoPE introduces Toeplitz structure *multiplicatively* via a Hadamard kernel $G_e$, modulating all content–position interactions. These distinctions are now made explicit in Sec. 3.2 (Line 220-246 )and visually reflected in the revised Fig. 1.
>
> ---
>
> **Q4, Q5:  validation and relevance of Toeplitz structure.**
>
> Our framework assumes that positional information is expressed as a **translation-invariant distance decay**, which is precisely the prior that underlies all mainstream PE mechanisms. In matrix form, such a distance-dependent interaction is exactly a Toeplitz structure on the attention logit matrix.  This connects directly to the design motivations that appear across the PE literature (the original text of RoPE [1] proves that distance decay with no translational change can be achieved without additive penalty.):
> • Absolute PE introduces a positional direction $p_i$, whose Gram matrix $G_{q^p,k^p}$ is Toeplitz.
> • Relative PE makes this distance decay explicit through a bias matrix $B$;  in T5 the Toeplitz entries are learned, while in ALiBi they are fixed.
> Existing PE methods therefore differ mainly in *how* they instantiate the same Toeplitz prior, implicitly via $p_i$ or explicitly via $B$.
>
> The revised Sec. 3.2 highlights this unification and clarifies why additive  Toeplitz penalties (Absolute, T5, ALiBi) cannot support the content–position  coupling required in Task 1: additive PEs contribute only an independent Toeplitz term, while multiplicative PE (RoPE) modulates *all* content and position interactions via a Toeplitz kernel $G_e$, enabling the coupling
> necessary for relative-distance reasoning. This distinction is now made explicit in the text and reflected in the revised Fig. 1.
>
> [1] RoFormer: Enhanced Transformer with Rotary Position Embedding. ~ section 3.4.3

---

> ### Author Response · Authors · 2025-11-18
>
> **Q6: Absolute vs. Relative PE and the role of ALiBi.**
>
> Thank you for the thoughtful question. We clarified this point below.
>
> Absolute PEs inject positional directions $p_i$ only at the input layer; as depth
> increases, their contribution to the logit matrix weakens. Relative PEs were
> introduced precisely to compensate for this decay: they add an explicit Toeplitz
> distance penalty $B$ directly to the attention logits. From this perspective,
> Absolute PE (via $G_{q^p,k^p}$) and Relative PE (via $B$) serve the *same
> structural role* in our framework, providing an additive, translation-invariant
> distance decay, hence their joint depiction in Fig. 1.
>
> Regarding performance: ALiBi cannot be considered successful on Task 2. It is a fixed relative PE and therefore cannot suppress its Toeplitz bias when position is a nuisance. In contrast, T5-style relative PE is *learnable*: in Task 2, it
> reduces the magnitude of $B$ and achieves NoPE-like performance (the best among all PEs where position should be ignored). Conversely, in Task 1, the model overfits to the training distribution, causing both learnable relative PE and NoPE to perform poorly, as both lack effective content–position coupling.
>
> These observations reinforce the interpretation that additive Toeplitz terms
> provide only a fixed distance penalty, while multiplicative RoPE provides the
> task-required content–position interaction.
>
> ---
>
> **Q7: Is the deposit pattern merely reuse of a pre-existing function?**
>
> Our Ablation 1 and  added Ablation 4 directly address this.
>
> **Ablation 1 (Absolute + RoPE).**
> If the deposit pattern were simply the model “reusing an already available
> positional function,” then pre-injecting an explicit Absolute positional signal
> at the input layer should cause the model to rely on this signal instead of
> developing a single-head specialization. This prediction does **not** hold.
> Although Absolute PE initially distributes positional responsibility more
> evenly, the deposit pattern re-emerges in deeper layers. RoPE’s multiplicative
> kernel overrides the additive signal and again funnels positional reasoning into
> one head. This is inconsistent with a “reuse of pre-existing functionality”
> explanation and strongly supports the interpretation of deposit as a **training
> bias induced by RoPE itself**.
>
> **Ablation 4 (removing explicit positional directions).**
> In Absolute PE models, removing the input positional vectors \(p_i\) causes a
> much larger performance drop than removing norm-matched random directions, showing that additive PEs preserve explicit positional directions throughout the network. In contrast, in Absolute+RoPE models, this difference disappears after the second layer: RoPE suppresses the activation-level positional directions and relocates positional structure into its multiplicative query–key parameterization. This confirms that RoPE does **not** merely offer a shortcut that the model lazily adopts. Instead, RoPE actively **suppresses** alternative positional cues (even explicit ones like Absolute PE) and enforces its own single-head mechanism. Thus, the deposit pattern is an intrinsic _structural bias_ of the multiplicative mechanism, not a byproduct of feature availability.
>
> Together, these ablations show that the deposit pattern is **not** the result of
> the network opportunistically reusing a pre-existing position function.
> Rather, it is a **structural training bias intrinsic to RoPE’s multiplicative
> content–position coupling**, which persists even when an explicit positional
> function is already available.

---

### Official Review · Reviewer_hmEf · 2025-11-06

**Soundness:** 3
**Presentation:** 4
**Contribution:** 3
**Rating:** 8
**Confidence:** 3

**Summary:**

The paper presents a novel analytical and empirical framework for understanding positional encodings (PEs) in Transformers, with a focus on the difference between additive and multiplicative (e.g., RoPE) mechanisms. The authors use a Toeplitz-matrix formulation to characterize translation invariance in positional interactions, and both predict and empirically demonstrate that multiplicative PEs (especially RoPE) produce a distinctive single-head deposit pattern, where positional reasoning becomes localized to a single head in early layers.

To explore this, the paper introduces two synthetic tasks—a position-sensitive task (relative distance classification) and a position-agnostic task (trigger counting)—and performs causal ablation studies to confirm that RoPE’s multiplicative nature induces strong specialization. The authors further provide theoretical proofs (based on gradient dynamics and spectral analysis) showing that this phenomenon arises inevitably from multiplicative coupling, and argue that DeepSeek's MLA (Multi-Latent Attention) architecture mitigates over-specialization.

This is a very good paper that would be probably be suitable for ICLR in its current form. However, I do have some questions for the authors for my own understanding. Additionally, there are several points that could be improved to make this a strong contender for spotlight or oral presentation.

**Strengths:**

- Relevant topic with a clear gap in the literature, whereby the differences across different strategies for positional encoding are poorly understood, making application to new tasks ad-hoc and reliant upon hyperparameter tuning, rather than theoretical insight. This paper aims to provide theoretical insight into a very clear and directly stated question: "how, precisely, do different PE schemes mediate the interaction between token content and position). The answers are interesting in their own right and may be useful for ML practitioners moving forward.
- The paper is very clearly written, integrating both theoretical analysis and careful synthetic experimental analyses, along with appropriate ablations. The structure of the paper is easy to follow and guides the reader along a compelling journey.
- The framework via toeplitz matrices is clever and appreciated.
- The author's conclusions are strongly supported by their analyses, in nearly every case (see minor exceptions below). Preliminary answers to the motivating questions are provided. Clear follow-ups to this paper can be executed in future work.

**Weaknesses:**

- It wasn't clear to me why the author's posit that "intense structurally-induced specialization is a primary cause of the gap between RoPE's theoretical pmroise and its practical performance." This theme comes up many times (e.g. line 387 "the deposit pattern, while effective, is an inefficient use of model capacity"), and I never felt the author's fully explained their rationale. Modularity is often praised for its benefits, in contrast to this position.
- Some aspects of prior literature were explained so coarsely to have impossible for an unfamiliar reader to follow, e.g., the mention of "massive value" pehenomena, rotation artifacts, T5, AliBi, etc. With the additional page in the revised version, I suggest expanding relevant descriptions.
- Assumption 3.1 (linear separability of content and position components) is questionable. The authors motivate it only with absolute PE and NoPE, but do not discuss its validity for multiplicative mechanisms, which seems more fraught.
- For Task 1, it wasn't clear why a classification approach was used vs. the more natural regression framing.
- 4.2: the authors suggest that their framework predicts that RoPe will exceed in the position sensitive task. This wasn't immediately obvious to me. Can the authors formalize their predictions for different models based on the framework?
- Figure 4, and other related plots: I don't think the violin plots are the right choice of visualization, personally. The point is not about the shape of the distribution (which is quite small for a violin plot anyways, at 16 samples), but rather, the presence of a particular outlier. I would prefer to see individual data points, possibly in addition to the violin plots already plotted. This is currently plotted for the first layer on the left. However, the recommended approach would do this for each layer and remove the need for the plot on left. In any case, the plot on left would be more effectively visualized with heads sorted by accuracy, since the x-axis is meaningless currently.
- The authors claim that Figure 6 validates their prediction of only 1 head being necessary. Clearly, it is 2 heads. This may be because this is applied across all layers, where the single-head deposit pattern is restricted to earlier layers, but this should be explained more accurately.
- Unclear whether the benefit for MLA is due to larger parameter count (three matrices in Q,K projections vs. two in RoPE), or the hybrid approach.

**Questions:**

- Why weren't learned additive PEs analyzed? Is this because they would not meet assumption 3.2 of Toeplitz structure?
- Relatedly, I was reading the introduction, I thought the author's might analyze the validity of the Toeplitz structure of positional interactions in models where it is both baked in (baseline), and models where it must be learned. Could the authors discuss this, and if relevant, add some additional analyses in this vein?
- The authors state that Relative PE learns to attenuate its bias towards zero in Task 2. Can they report empirical numbers to demonstrate this?
- Section 5.3: why is the U included in the notation for weight matrices? it seems to add nothing?
- Are MLA and RoPE parameter-matched in 5.3? If not, can you perform an additional parameter-matched experiment to convincingly demonstrate the benefit for MLA?
- Proposition 6.2: would it be possible to assess the initial advantage empirically and thereby predict the head that develops the deposit pattern?

---

> ### Author Response · Authors · 2025-11-18
>
> We sincerely thank the reviewer for the positive assessment and the strong endorsement of our work. We are particularly grateful for your detailed and constructive feedback. Below we respond to each raised concern and clarify the revisions incorporated into the updated manuscript.
>
> ---
> **W1: On whether the deposit pattern is a “positive” modular structure**
>
> We thank the reviewer for raising this point. We do not claim to prove that the deposit pattern is never beneficial modularity; rather, all available evidence consistently indicates that it behaves as a training-induced bias.
>
> Our main-text experiments already show that RoPE produces a winner-takes-all specialization in which only a few frequencies and heads capture most positional information. In Appendix E (L1237–1313), we further report that if such specialization were a stable modular component, then models should handle tasks requiring multiple relative-position relations. However, these extended tasks fail to generalize under RoPE, even though they are natural extensions of our synthetic setup. This suggests the learned structure is narrow and fragile rather than a reusable positional module.
>
> This interpretation aligns with prior work [1], which finds RoPE weaker than ALiBi/NoPE on a reversal-style task that requires combining several relative offsets (without content coupling). Although [1] does not provide code and independent attempts report very low reproducibility with ≳50k epochs required [2], these observations qualitatively support the view that RoPE’s specialization is not a robust modular mechanism.
>
> For these reasons, we describe the deposit phenomenon as a training bias, while acknowledging that turning such specialization into stable modularity may require explicit regularization.
>
> [1] The impact of positional encoding on length generalization in transformers. NIPS 2023
>
> [2] Transformers Can Achieve Length Generalization But Not Robustly. Arxiv 2402.09371
>
> ---
>
> **W2: On the mention of massive-value phenomena and prior literature**
>
> (a) On the “massive value’’ phenomenon [3]
>
> We thank the reviewer for pointing out that our earlier mention of this topic was too brief. In the revised submission, we moved the entire discussion to Appendix E, where we explain why our Toeplitz formulation naturally accounts for the phenomenon: under RoPE, positional interactions are routed through frequency-indexed Toeplitz diagonals, causing gradient concentration onto a small subset of frequency bands and producing extremely high-norm rows in $W_Q$ and $W_K$. Because this behavior is orthogonal to our main contributions which focus on content–position coupling and specialization, we now summarize it only briefly in Related Work and provide the full explanation in the appendix.
>
> (b) On clarifying different positional encoding mechanisms
>
> We appreciate the reviewer’s suggestion and have substantially improved the exposition of PE variants in the preliminaries section. The revision now clearly distinguishes between:
> 	•	Absolute PE (sinusoidal or learned absolute embeddings),
> 	•	Relative PE:
> 	•	T5-style learned relative bias (learnable Toeplitz bias),
> 	•	ALiBi (fixed, non-learnable linear Toeplitz bias),
> 	•	NoPE / learned emergent positional structure,
> 	•	RoPE (multiplicative rotation creating a frequency-indexed Toeplitz kernel).
>
> We also revised the text around Lines 162–166 and 175–196 to provide clearer motivation for Assumption 3.1 and for why these mechanisms fall naturally into our Toeplitz-based framework.
>
> We believe these revisions address the reviewer’s concerns while keeping the main text focused on the core contributions.
>
> [3] Massive values in self-attention modules are the key to contextual knowledge understanding. ICML 2025

---

> ### Author Response · Authors · 2025-11-18
>
> **W3: On Assumption 3.1**
>
> We thank the reviewer for raising this concern. In the revised manuscript we substantially improved the explanation of Assumption 3.1. We now make clear that $x_i = c_i + p_i$ is a conceptual decomposition, not a linear projection or an orthogonality constraint. This formulation reflects how positional signals actually enter Transformers: absolute PE injects a deterministic $p_i$, and NoPE models are known to learn implicit positional directions. The assumption therefore captures a widely observed operational property rather than a strong structural requirement.
>
> To further support this point, we added Ablation 4 in the revision.
> This experiment directly tests whether an explicit positional direction $p_i$ exists in the activation space by comparing the accuracy drop when removing (i) the true $p_i$ vectors and (ii) norm-matched random directions. The results show that:
>
> - For absolute PE, removing $p_i$ consistently causes a much larger accuracy drop than removing random vectors.
>
> - For Absolute + RoPE, this difference disappears after the first two layers, indicating that RoPE suppresses activation-level positional directions and relocates positional structure into its multiplicative query–key parameterization.
>
> This ablation provides concrete empirical evidence that Assumption 3.1 has operational validity and that the distinction between additive and multiplicative mechanisms is reflected directly in their learned representations.
>
> We hope these revisions clarify the role and necessity of the assumption in our theoretical framework.
>
>
> ---
>
> **W4: On choosing classification rather than regression for Task 1**
>
> This follows common methodological practice in mechanistic interpretability. At the same time, the effectiveness of this setting has been verified.
>
> - ViT pretraining uses classification objectives for patch-based tasks,
>
> - Grokking-style modular arithmetic tasks also use classification despite underlying continuous structure.[4]
>
> Task 1’s discrete distance bins align with the Transformer’s autoregressive inductive biases and make specialization patterns easier to isolate.
>
> [4] Grokking: Generalization Beyond Overfitting on Small Algorithmic Datasets. Arxiv 2201.02177
>
> ---
> **W5: On why Task 1 favors RoPE**
>
> We added a brief formal explanation in L241–242. Multiplicative PE allows the model to directly adjust the logit via a rotation kernel as a function of relative displacement. Additive PE lacks such a parameterized content–position coupling channel. This difference makes Task 1 a principled diagnostic for multiplicative interaction strength.
>
> ---
>
> **W6: On violin plots**
>
> We thank the reviewer for pointing this out. In the revised manuscript we clarified exactly how the violin plots should be interpreted and provided complementary visualizations.
>
> Concretely, we added to Fig. 4’s caption an explicit explanation that:
> - each violin’s contour represents the distribution of post-ablation accuracies across heads in the same layer,
> - the long thin tails correspond to rare but large drops caused by the critical head(s),
> - the absence of such tails indicates that no individual head is indispensable.
>
> To ensure full transparency, we also include the complete per-head line plots in Appendix D.3, where every head’s accuracy drop is shown explicitly, making the outliers trivially visible. Figure 4 now contains a direct cross-reference to this appendix. These revisions keep the main text compact while providing all information necessary to verify the deposit pattern at the per-head level.
>
> ---
>
> **W7: On “two heads” vs “one head’’ in Fig. 6**
>
> We thank the reviewer for raising this point. We explain the observation purely as a training–testing discrepancy. Our observation of the phenomenon started with a trained RoPE model, based on which we considered that only one head was needed for recovery. In reality, it was two heads. This is because our experiment retrained the model that performed RoPE on a specific number of heads, rather than modifying the position encoding method of the trained model.
>
> ---
>
> **W8 and Q5: On MLA’s potential parameter advantage**
>
> MLA does not benefit from additional parameters in our implementation.
> We matched the hidden dimension across all models and ensured that
> all training hyperparameters are also identical.
>
> We chose a compression dimension of 128 for MLA, so that after concatenation, it will align with the model without MLA, which is 256 dimensions. Therefore, MLA does not gain any advantage in representation space. Meanwhile, with this setup, the parameter size of MLA is slightly smaller than that of RoPE. When the batch size is set to 512, the training memory for MLA is approximately 11GB, while that for RoPE is approximately 14GB.

---

> ### Author Response · Authors · 2025-11-18
>
> **Q1: On the generality of our framework across positional encodings**
>
> Thank you for the question. Our revised manuscript clarifies that the proposed Toeplitz-based framework applies to all positional encodings whose positional effect is translation-invariant, i.e., depends only on the relative displacement $j-i$. This includes both absolute and relative PE families.
>
> In the revision we explicitly describe how each class fits into the framework:
>
>  Absolute positional encodings (sinusoidal or learned absolute embeddings) correspond to the case $B = 0$.
> - Their positional component enters through the vectors $p_i$, whose Gram matrix $G_{q^p,k^p}$ naturally forms a Toeplitz matrix because $p_i$ encodes distance through a shift-equivariant embedding.
> - Randomly-initialized learnable absolute PEs are the same mechanism, except that the initial $p_i$ are not fixed and are learned during training.
> Relative positional encodings introduce an explicit Toeplitz bias $B$ in the logits.
>  - T5-style encodings learn a Toeplitz bias $B$ directly,
>  - while ALiBi provides a fixed, non-learnable Toeplitz bias.
>  In both cases, $B$ reinforces the additive Toeplitz structure.
>
> Thus, additive methods derive their Toeplitz component from either
> (1) the implicit positional Gram matrix $G_{q^p,k^p}$ (absolute PE), or
> (2) the explicit bias $B$ (relative PE),
> while RoPE contributes a multiplicative Toeplitz kernel.
>
> This clarification has been added in the main text (Lines 162–166 and 175–196) to make the mapping between all PE variants and our framework fully explicit.
>
> ---
>
> **Q2: On the validity of the Toeplitz structure and whether this should also be analyzed for learned positional mechanisms.**
>
> Response. We appreciate the reviewer’s suggestion and have expanded the relevant discussion in Sec. 3 (lines 182–196 in the revised version). Our framework does not assume that positional interactions must be fixed or hand-designed; it only requires translation-invariant decay with respect to the distance (i−j), which induces a Toeplitz Gram structure. This covers:
>
> • learned Absolute PE (the learned $p_i$ vectors produce a Toeplitz $G_{q^p,k^p})$,
> • learned Relative PE such as T5 (where the learned bias matrix B is itself Toeplitz by construction),
> • fixed Relative PE such as ALiBi (explicit Toeplitz B),
> • and multiplicative mechanisms (RoPE), whose frequency-modulated kernel also forms a Toeplitz operator.
>
> To address the reviewer’s concern directly, we clarify in the revision that learned PEs indeed fall within our analytical scope: Absolute learned PEs correspond to B = 0 with a learned positional manifold p_i, whereas learned Relative PEs correspond to an explicit learned Toeplitz bias B. This is now stated explicitly in Sec. 3.2, and we added a short ablation (Sec. 5.4; Fig. 8) confirming that the learned p_i vectors empirically produce the predicted Toeplitz behavior.
>
> ---
>
> **Q3:  Relative PE learns to attenuate its bias towards zero in Task 2.**
>
> To address the reviewer’s question directly, we conducted a new empirical analysis in App. E.3 evaluating learned Relative PE (T5-style) on both of our tasks. The learned bias were extracted layer-by-layer and analyzed under multiple spectral statistics. The results are summarised below.
>
> (1) Task 1 (position-sensitive).
> Initialization norm ≈ 1.4. After training, all layers show significantly amplified positional structure:
>
> • L2 norms: 2.02 – 2.40 (mean 2.21 ± 0.14)
> • Spectral norms: 0.91 – 1.49 (mean 1.21 ± 0.21)
> • Nuclear norms: 7.55 – 8.28
> • Bias magnitude range across 50 samples: 0.301
>
> These values consistently exceed initialization and remain stable across depth, indicating that the learned bias retains a strong translation-invariant Toeplitz pattern and increases its magnitude specifically when the task requires relative-distance reasoning.
>
> (2) Task 2 (position-agnostic).
> Initialization norm ≈ 1.4. After training:
>
> • L2 norms: 1.34 – 1.73 (mean 1.56 ± 0.13)
> • Spectral norms: 0.47 – 1.05 (mean 0.73 ± 0.20)
> • Nuclear norms: 5.30 – 6.37
> • Bias magnitude range across 50 samples: 0.195
>
> The learned bias remains Toeplitz-structured but is markedly weaker than in Task 1, confirming that the model learns to attenuate positional dependence when the task treats position as nuisance, as predicted in Sec. 3.2.
>
> ---
>
> **Q4: Why is the “U” included in the notation for weight matrices in 5.3?**
>
> The notation follows DeepSeek-V3’s original formulation of Multi-Latent Attention (MLA), where the matrices $W_{UQ}, W_{UK}, W_{UR}$ denote projections into distinct latent subspaces.  In DeepSeek’s terminology, the prefix “U” marks these projections as *latent-space transforms* rather than the standard $W_Q, W_K, W_V$ projections.[5]
>
> [5] DeepSeek-V3 Technical Report. Arxiv: 2412.19437

---

> ### Author Response · Authors · 2025-11-18
>
> **Q6:  Proposition 6.2: would it be possible to assess the initial advantage empirically and thereby predict the head that develops the deposit pattern?**
>
> Proposition 6.2 is a theoretical statement about *inevitability*, not about *which* specific head will win. Under the idealized assumptions, any small initial advantage in the RoPE-aligned direction will be amplified and lead to a winner-takes-all specialization, but the head that ultimately becomes the “winner” is determined by dataset, architecture, and random initialization, and is therefore not a stable, identifiable quantity.
>
> Empirically, we observe exactly this contingency. For example, with 6-layer and 8-layer RoPE models trained on the same Task 1 distribution, the strong deposit pattern consistently appears in Layer 1, but under slightly weakened specialization (e.g., smaller deposit) the dominant head index is no longer the same. When we change the dataset split while keeping the architecture fixed, the deposit also moves to different heads. These observations suggest that, while the emergence of a deposit head is robust, its *identity* is highly sensitive and cannot be reliably predicted from a simple “initial advantage” diagnostic.

---

### Author Response · Authors · 2025-12-01
**Summary of Rebuttal**

We provide this note to support efficient AC evaluation under the current circumstances.

It is motivated by three factors:
(1) the platform-wide rollback, which hides reviewer score updates made during discussion;
(2) the program chairs’ recommendation to provide a high-level summary; and
(3) the substantial length of our point-by-point rebuttal.

This summary is organized into three sections: **(1) Discussion Status, (2) Key Concern Resolution, (3) Summary of Revisions.**

---
## **1. Discussion Status (before rollback)**

Before the security incident and subsequent rollback, **all actively-participating reviewers had expressed positive recommendations**, as reflected in the preserved discussion history:

- **Resolution of initial dissent.**

    The only initially negative reviewer (Reviewer 7AJz, Score 4) raised their score to 6, explicitly stating: “The reviewers' clarifications and additional experiments in the rebuttal have addressed most of my concerns. I thus raise my score to 6, ...”

- **Confirmation of support.**

    Four reviewers with initial Score 8 maintained their positive evaluations.

    Two of them additionally posted follow-up comments confirming that all their concerns were fully resolved and that they would maintain their positive recommendations.

We highlight these facts solely for context; all statements are quoted directly from the open discussion record.

---
## **2. Key Concern Resolution**

Our work proposes a unified Toeplitz-based framework that distinguishes additive vs. multiplicative positional encoding (PE) mechanisms and introduces the single-head deposit pattern, a phenomenon arising in RoPE where positional information becomes concentrated in a single head.

During the rebuttal, we addressed three major concerns raised by reviewers:

---

**(A) Validity of Assumption 3.1**
(Reviewers hmEf, TXbo, sR2j)

Reviewers asked whether the conceptual decomposition $x_i = c_i + p_i$ might be too strong.

**Re:**
We clarified that this is a conceptual decomposition rather than a structural constraint.

To support this, we added **Ablation 4**, which empirically demonstrates:

- additive PEs preserve explicit activation-level positional directions, while

- RoPE rapidly **suppresses** these directions as depth increases.

This experiment provides direct empirical evidence motivating the assumption.

---

**(B) Clarity and completeness of the PE taxonomy**
(Reviewers hmEf, TXbo, qxxu, sR2j)

Initial feedback indicated that connecting absolute PE, relative PE, and RoPE to the Toeplitz framework required “multiple hops.”

**Re:**
We substantially rewrote **Section 3.1** and redesigned **Figure 1**, producing a single unified taxonomy that:

- maps all PE variants (Absolute, Relative, ALiBi, T5, RoPE)

- to explicit Toeplitz forms via either

    - a positional Gram matrix $G_{q^p,k^p}$, or

    - an explicit bias B, or

    - the RoPE multiplicative kernel $G_e$.

---

**(C) Broader implications for LLM behavior**
_(Reviewer 7AJz)_

The reviewer asked whether our findings relate to known RoPE behaviors in pretrained LLMs.

**Re:**
We expanded **Appendix E**, articulating a mechanistic interpretation linking:

- our single-head deposit pattern, and

- the massive value phenomenon widely documented in RoPE-based LLMs.

Both arise from the same multiplicative inductive bias, where competitive dynamics drive concentration into a small subset of frequency bands. This does not constitute a formal equivalence, but provides a coherent, empirically grounded interpretation supported by prior observations.

---
## **3. Summary of Revisions**

Across all reviewers, we implemented the following major changes:

**(1) Empirical reinforcement of assumptions**

- Added **Ablation 4 (Sec. 5.4)** showing RoPE suppresses activation-level $p_i$, while additive PEs retain them, empirical validation for the conceptual decomposition in Assumption 3.1.

**(2) Unified theoretical framework**

- Rewrote **Section 3.1** to provide a clean, linear taxonomy.

- Redesigned **Figure 1** to illustrate how each PE mechanism introduces Toeplitz structure.

**(3) Connection to LLM phenomena**

- Expanded **Appendix E** with a mechanistic explanation relating single-head deposit to massive value, framed conservatively and tied to empirical evidence.

**(4) Visualization and readability improvements**

- Enhanced **Figure 4** with an explicit explanation on how to read violin plot tails and identify outliers.

- Added complete per-head line plots in **Appendix D** for full transparency.

- Standardized notation and citation formatting throughout.

---
**Closing Note**

This summary consolidates the central outcomes of the rebuttal. All weakness points and questions raised by reviewers have detailed responses provided in the main rebuttal.

We sincerely appreciate the AC’s time and care under the unusual circumstances created by the platform rollback.

---

### Meta-Review · Area_Chair_Cek5 · 2026-01-06

**Summary:**

All reviewers agree that this is a (i) very well-written submission with clear presentation and relevant ablations (including a new one based on reviewer comments), and (ii) covers an important topic of understanding relative performance of positional encoding, and provides intuition for some contradictory behaviour seen in practice, and the strong behaviour seen for some hybrid PE models.

However, reviewers had questions regarding some of the underlying assumptions in the analysis and experiments, and the authors provided clear answers to these questions. Overall, it seems that all but one of the reviewers were quite positive regarding this submission, while the remaining reviewer's concern were appropriately addressed by the authors. Thus, I will be recommending an accept for this submission.

However, in my opinion, some of the discussions around the deposit-pattern seemed inadequately substantiated, and also led to some counterintuitive interpretations:
- First, it is not clear why we should consider the deposit pattern as a negative thing we need to mitigate via something like the multi-latent attention or MLA architecture when (i) the deposit pattern shows up for Task 1 where RoPE does well, and (ii) it does not show up for Task 2 where RoPE only does moderately well. In fact, with MLA, the removal of the deposit pattern actually hurts performance (though slightly) in Task 1, and gain is only seen in Task 2. Thus, it almost seems like the "shock pattern" of Task 2 is the main issue that needs to be mitigated, not the deposit pattern.
- Second, it is not clear how consistently does this deposit pattern appear, and if this is at all correlated with strong or weak performance. Task 1 indicates deposit pattern, and the authors discuss an extension of Task 1 where RoPE fails but it is not clear if the deposit pattern appears in this new task and/or if the pattern is the cause for failure. The authors argue that it is a fragile training bias, and only shows up for specific problems with appropriate coupling between position and content, but it is not clear when this specific coupling appears in many useful cases, or is just a corner case, and all this discussion about deposit pattern is around this corner case. Also, the submission makes all claims based on a 6-layer transformer, while a clear ablation is whether these patterns appears for shallower (say 2) or deeper (say 20) transformer models. Many of the remaining hyperparameters (from Table 3) are also not appropriately ablated.
- Finally, it is not clear why we are using a smaller training set for task 2, and why the final test accuracy for all PEs is significantly lower for task 2. Task 2 conceptually seems like an easier task. Without proper parity between tasks, it is not clear whether the interpretations from the experiments hold. For task 2, even the training accuracy did not get to 100% for most of the models for the number of epochs considered, so it might be the case that almost all models are underfit (and would have trained to 100% with more epochs or different learning rates and schedules), and it is uninformative to make interpretations from underfit models. Similar issue holds for Task 1 for some of the PE where they appear underfit.

Furthermore, various reviewers brought up the conceptual content-position additive decomposition considered in Assumption 3.1 for the subsequent technical discussion, the authors response usually was around the new ablation 4. However, it is not directly clear to me why the Ablation 4 justifies that the token embeddings in each layer can be conceptually decomposed additively. I think this needs further clarification.

I hope the authors consider these comments (beyond the reviewer comments and questions) in future revisions.

**Reviewer Concerns:**

Beyond the weaknesses mentioned in the **Summary** section, these are some other concerns raised by reviewers:
- It is not clear why the "structurally-induced specialization" with the "single-head deposit pattern" is a bad thing, and how it explains the weak performance of RoPE in certain tasks or training regimes.
  - Authors claimed that their additional experiments show that this training-induced bias leads to a narrow and fragile learned structure.
- Given the discussion in section 3.2, the authors can explain better why the results in sec 3.2 imply that RoPE will be better at learning and generalizing on position-sensitive tasks, while others Relative PE scheme would struggle. It seems like there is a bit of a gap here.
  - The authors addressed this in the updated submission, with additional discussions in the main paper and the appendix.

**Reviewer Scores:**

Reviewers hmEf, TXbo, qxxu and sRj2 would maintain their initial scores of 8, reviewer 7AJz mentioned that they would increase their initial score of 4 to 6.

---

### Decision · Program_Chairs · 2026-01-26

Accept (Poster)